# Stretchable hydrogels with low hysteresis and anti-fatigue fracture based on polyprotein cross-linkers

Hai Lei[1,2,7], Liang Dong[1,7], Ying Li[1,3], Junsheng Zhang[1], Huiyan Chen[1], Junhua Wu[4], Yu Zhang[1], Qiyang Fan[5,6], Bin Xue[1], Meng Qin[1], Bin Chen[5,6], Yi Cao [1,2✉] & Wei Wang [1]

Hydrogel-based devices are widely used as flexible electronics, biosensors, soft robots, and intelligent human-machine interfaces. In these applications, high stretchability, low hysteresis, and anti-fatigue fracture are essential but can be rarely met in the same hydrogels simultaneously. Here, we demonstrate a hydrogel design using tandem-repeat proteins as the cross-linkers and random coiled polymers as the percolating network. Such a design allows the polyprotein cross-linkers only to experience considerable forces at the fracture zone and unfold to prevent crack propagation. Thus, we are able to decouple the hysteresis-toughness correlation and create hydrogels of high stretchability (~1100%), low hysteresis (< 5%), and high fracture toughness (~900 J m$^{-2}$). Moreover, the hydrogels show a high fatigue threshold of ~126 J m$^{-2}$ and can undergo 5000 load-unload cycles up to 500% strain without noticeable mechanical changes. Our study provides a general route to decouple network elasticity and local mechanical response in synthetic hydrogels.

[1] Collaborative Innovation Center of Advanced Microstructures, National Laboratory of Solid State Microstructure, Department of Physics, Nanjing University, Nanjing 210093, China. [2] Chemistry and Biomedicine Innovation Center, Nanjing University, Nanjing 210093, China. [3] Institute of Advanced Materials and Flexible Electronics (IAMFE), School of Chemistry and Materials Science, Nanjing University of Information Science and Technology, Nanjing 210044, China. [4] Jiangsu Key Laboratory of Molecular Medicine, Medical School, Nanjing University, Nanjing 210093, China. [5] Department of Engineering Mechanics, Zhejiang University, Hangzhou 310027, China. [6] Key Laboratory of Soft Machines and Smart Devices of Zhejiang Province, Hangzhou 310027, China. [7]These authors contributed equally: Hai Lei, Liang Dong. ✉email: caoyi@nju.edu.cn

The development of soft stretchable materials, including elastomers and gels[1,2], enables the fast-growing fields of flexible electronics[3,4], tissue engineering scaffolds[5,6], and smart drug delivery systems[7]. In many of these applications, high stretchability, fracture toughness, and low hysteresis are the prerequisite; yet most hydrogels are too fragile to tolerate cyclic mechanical loads. Single-network hydrogels lack a mechanism to prevent crack propagation and facilitate energy dissipation. Therefore, they are intrinsically weak and prone to fracture. On the other hand, double-network hydrogels are famous for their high stretchability and toughness due to the presence of a sacrificial network that can be fractured to dissipate mechanical energy[8–13]. However, they inevitably show obvious hysteresis in the stretching–relaxation cycles, making them unsuitable for applications requiring dynamic mechanical loads. Moreover, recent studies have showed that double-network hydrogels are unable to prevent facture propagation at the strain limit in that the sacrificial network has already been ruptured before reaching the fracture point[14,15]. As such, the fracture energy of tough double-network hydrogels is comparable to the intrinsic fracture energy of single-network hydrogels[14,15]. Recently, Zhao and coworkers developed a way to engineer anti-fatigue fracture hydrogels by inducing crystalline phases to prevent crack propagation[16]. Despite that the fatigue threshold was significantly improved, the hysteresis may remain an issue. The stretchability, toughness, hysteresis, and anti-fatigue fracture are all the results of energy dissipation but under different conditions. The seemingly conflict requirements of low hysteresis, high toughness, and fracture resistance make it challenging to design hydrogels combining these mechanical properties.

Unlike synthetic hydrogels, many biological tissues, such as muscle[17,18] and cartilage[19], show exceptional mechanical properties and can survive under millions of mechanical cycles in their life span. In many of these tissues, nature has evolved a special class of elastomeric proteins made of tandem repeats of folded protein domains to function as cross-linkers for loosely packed proteineous fibers[20–22]. These elastomeric proteins can unfold to dissipate mechanical load and quickly refold to recover their original mechanical properties. Inspired by this design, Li and others have pioneered the use of folded protein domains as the building blocks for engineering synthetic hydrogels with tailored mechanical properties[23–25]. Despite great success in these studies to partially mimic the mechanical response of biological tissues, most engineered hydrogels were still limited by obvious hysteresis, poor stretchability and low fracture thresholds. It remains largely unexplored to engineer stretchable, low hysteresis, and anti-fatigue hydrogels using biomimetic approaches.

Recently, theoretical and experimental studies on extracellular matrix and cell cytoskeleton have suggested that the network structure, not just the protein composites, plays an important role in their mechanical response[26–31]. Many unique mechanical responses[32], including active superelasticity[33] and high compressibility[34–36], are indeed stemmed from special combination of protein networks of different rigidities and interaction dynamics. Moreover, mechanical enhancement can be achieved in the hydrogels with only one polymer forming the percolating phase and the other as the inclusion or cross-linkers, to decouple the network elasticity with local mechanical response[37–39]. Inspired by these studies, we propose a network structure that is made of the unstructured polymers as the percolating phase and the polyproteins as the cross-linker to achieve combined low hysteresis and anti-fatigue fracture properties. This network structure is distinct from the network structures in which the folded protein domains are stringed together with the unstructured polymers to form the percolating phase[24]. In that case, the forces on folded proteins and the unstructured polymers are the same, which inevitably leads to protein unfolding during stretching and introduces hysteresis[24].

In this work, we show that the macroscopic deformation of the hydrogel is mainly contributed by the extension of the percolating unstructured polymers but not the polyprotein cross-linkers, as the cross-linkers are mechanically bypassed. The forces only propagate to the polyprotein cross-linkers when the unstructured chains are considerably tightened. Even at large strains, the extension of the polyprotein cross-linkers is small. This prevents the protein domains from unfolding during stretching and allows them to remain folded at high strains. Therefore, the hydrogels show low hysteresis upon stretching. However, these folded protein domains can still be unfolded at the stress-concentrated crack area to efficiently prevent crack propagation, entailing the hydrogels high fatigue resistance. We anticipate that the design can result in hydrogels of low hysteresis, high strain limit, and anti-fatigue fracture properties.

## Results

**Design and engineering of the hydrogels**. The hydrogels were made of polyacrylamide (PAA) as the percolating phase and the polyprotein comprising of eight tandem repeats of GB1 (G8) as the cross-linker (Fig. 1a). GB1 is mechanically stable[40] and has been extensively used for engineering hydrogels with outstanding mechanical properties[24,25,41]. Unfolding of G8 gives rise to saw-tooth-like patterns with forces of ~200 pN at a pulling speed of 1600 nm s$^{-1}$ or staircase-like patterns with lifetimes of ~0.1 s at a constant pulling force of 150 pN (Supplementary Fig. 1). In order to integrate the polyprotein with PAA network, we used the well-established SNAP chemistry[42]. We flanked both ends of the G8 with SNAP protein (SNAP-G8-SNAP) and allowed it to react with O6-benzylguanine styrene (BS) to covalently link a vinyl group to each end of the polyprotein (Fig. 1a). The hydrogels were prepared by a one-pot free radical polymerization of acrylamide and BS-linked SNAP-G8-SNAP in the phosphate buffer saline (PBS) buffer for 30 min using lithium phenyl-2,4,6-trimethylbenzoylphosphinate (LAP) as the photo initiator (Fig. 1a). Then the hydrogels were dialyzed against PBS to completely remove all undesired byproducts or unreacted reactants. The hydrogels were named as PAA-G8 hydrogel. For comparison, we engineered hydrogels using bisacrylamide as the cross-linker to reveal the contribution of the polyprotein to the overall mechanical properties[43]. The hydrogels were named as PAA hydrogel. We also prepared hydrogels containing the same polyprotein but having different network topology. The hydrogels were made of BS-linked SNAP-G8-SNAP and four-armed thiol-terminated polyethylene glycol (4-armed-PEG-SH)[24], in which the covalent cross-links were formed through the thiol-ene reaction. In this design, the polyproteins were linked with unstructured PEG molecules as part of the percolating network and the corresponding hydrogels were named as PEG-G8 hydrogel. In the PAA-G8 hydrogel, the concentrations of SNAP-G8-SNAP and acrylamide were both 100 mg mL$^{-1}$. For comparison, the molar ratio of bisacrylamide in the PAA hydrogel was the same as SNAP-G8-SNAP in the PAA-G8 hydrogel, so that the two hydrogels have the same theoretical cross-linking density. In the PEG-G8 hydrogel, the molar ratio of SNAP-G8-SNAP and 4-armed-PEG-SH was 2:1 and the concentration of G8 was the same as that in the PAA-G8 hydrogel. Note that in the PEG-G8 hydrogel, not only the network structure but also the cross-linking densities were different.

We expected that the three hydrogels showed distinct mechanical response and crack propagation mechanism (Fig. 1b). The PAA hydrogel contains only a single polymer network and lacks a mechanism to dissipate mechanical energy and prevent

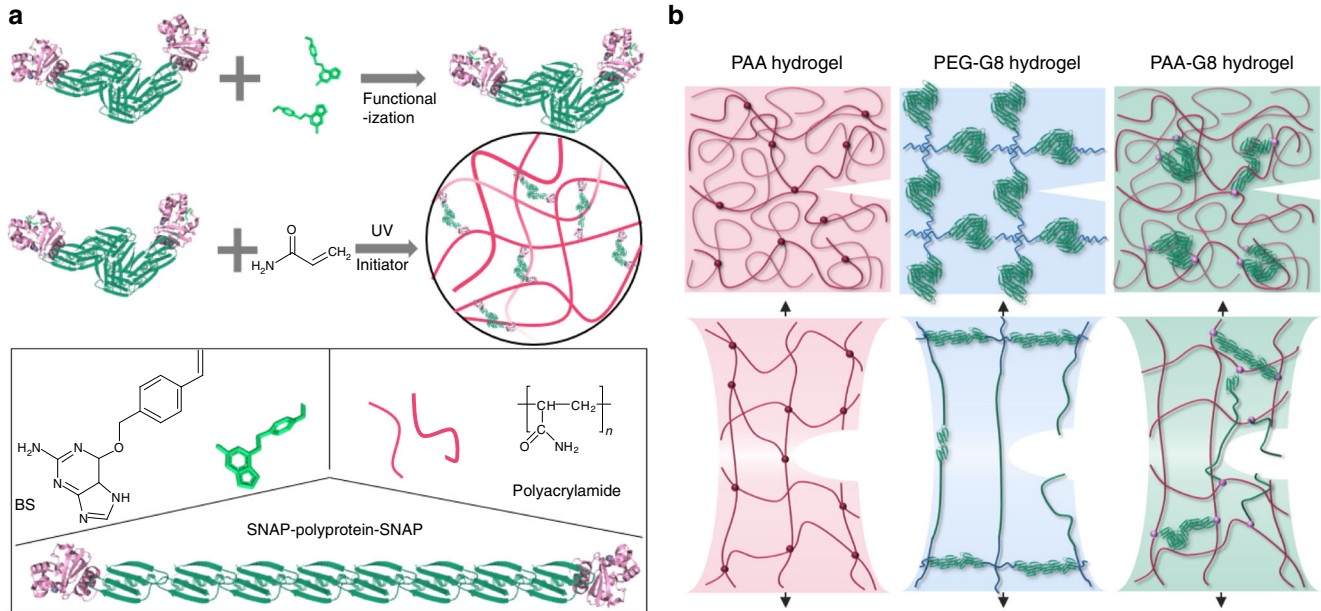

**Fig. 1 Design principle for the anti-fatigue-fracture hydrogels. a** Schematic illustration of the preparation of the PAA-G8 hydrogel. First, SNAP-G8-SNAP protein reacts with BS to covalently link a vinyl group to each end of the protein. Then the hydrogel is prepared by copolymerization of the polyprotein cross-linker and acrylamide using UV initiated free radical polymerization. **b** Illustration of the network structures of the PAA, PEG-G8, and PAA-G8 hydrogels before and after stretching to the critical crack propagation point. The green rods represent folded polyproteins and green lines represent unfolded proteins. In the PAA and PEG-G8 hydrogels, the fatigue threshold is determined by the energy required to fracture a single layer of polymer chains per unit area. In the PAA-G8 hydrogel, the fatigue crack propagation requires both the fracture of polymer chains and the unfolding of polyproteins at the single layer of the crack front.

crack propagation. The PEG-G8 hydrogel comprises a network of folded GB1 domains and PEG polymers[44]. Upon stretching, the GB1 domains unfold gradually to dissipate the mechanical load. Because the GB1 domains are already unfolded before reaching the critical crack propagation point, the crack propagation process is still dominated by the scission of single-layered polymer chains. However, in the PAA-G8 hydrogels, the macroscopic deformation of the hydrogels is mainly contributed by the extension of the percolating PAA phase but not the G8 cross-linkers, as the G8 cross-linkers are bypassed (Fig. 1b). The major cross-links that bear considerable forces are the entanglement points of the PAA chains. The forces only propagate to the G8 cross-linkers when the PAA chains are considerably tightened. Even at large stains, the extension of the G8 cross-linkers is small. The deformation of the network can only lead to the sharp increase of stretching forces on the G8 cross-linkers at extremely large strain beyond the fracture strain of the hydrogels. However, this strain limit can be met at the crack tip due to the stress concentration effect. The unfolding of GB1 then greatly prevents the crack propagation. Therefore, the hydrogel is expected to possess high stretchability, low-hysteresis, and anti-fatigue fracture properties.

**Mechanical characterization of the PAA-G8 hydrogels.** Next, we studied the mechanical properties of the three hydrogels using tensile test experiments. All mechanical tests were performed in air, at room temperature, using a tensile machine with a 10-N load cell. The rate of stretch was kept constant as 10 mm min$^{-1}$ if not otherwise mentioned. The PAA-G8 hydrogel was highly stretchable and can be extended 11 times its original length without break (Fig. 2a). Even with a precut notch, the hydrogel sample can still be stretched for more than 5.5 times without obvious increase of the notch length. The notch front became significantly blunt to prevent crack propagation. When the strain went beyond a threshold, the notch started to run very slowly,

which was distinct from the fast propagation of crack found in many single and double-network hydrogels (Fig. 2b and Supplementary Movie 1)[9,15,45].

The typical stress–strain curves of PAA, PEG-G8, and PAA-G8 hydrogels are shown in Fig. 2c–e, respectively. The PAA gel can be extended eight times its original length without rupture with a Young's modulus of 5 kPa at 5% strain. The PEG-G8 hydrogel can only be extended 1.8 times with a much higher Young's modulus of ~60 kPa. However, the PAA-G8 hydrogel can be stretched to more than 11 times its original length without rupture with a Young's modulus of ~12 kPa. The stretchability and Young's modulus are both associated with the cross-linking density of the hydrogels. It is interesting that the PAA-G8 hydrogel showed distinct rupture behavior. The PAA and PEG-G8 hydrogels ruptured rapidly within a second (Fig. 2c, d, inset). However, the PAA-G8 hydrogel ruptured at a much slow crack propagation speed. The stress dropped gradually, and the stress–strain curve showed a large fracture zone. We hypothesized that the presence of such fracture zone (Fig. 2e) was due to the unfolding of GB1 at the crack front, which greatly dissipated the fracture energy and delayed the fracture events.

To confirm that GB1 domains remained folded till the fracture point in the PAA-G8 hydrogel but not in the PEG-G8 hydrogel, we performed the load/unload cyclic test on all three hydrogels to different strains. Because the PAA hydrogels did not contain any folded proteins, the stretching–relaxation cycles showed no hysteresis (Fig. 2f). In contrast, the PEG-G8 hydrogels showed clear hysteresis which increased with the increase of strain (Fig. 2g). Even though GB1 can refold quickly, the unfolding and refolding of GB1 were irreversible in the PEG-G8 hydrogels. In the PAA-G8 hydrogels, the stretching–relaxation cycles did not show any hysteresis even at a strain of 1000% (Fig. 2h). If GB1 was unfolded during the stretching process, we would expect to see similar hysteresis as shown in the PEG-G8 hydrogels. The overlapping of the stretching and relaxation traces indicated that

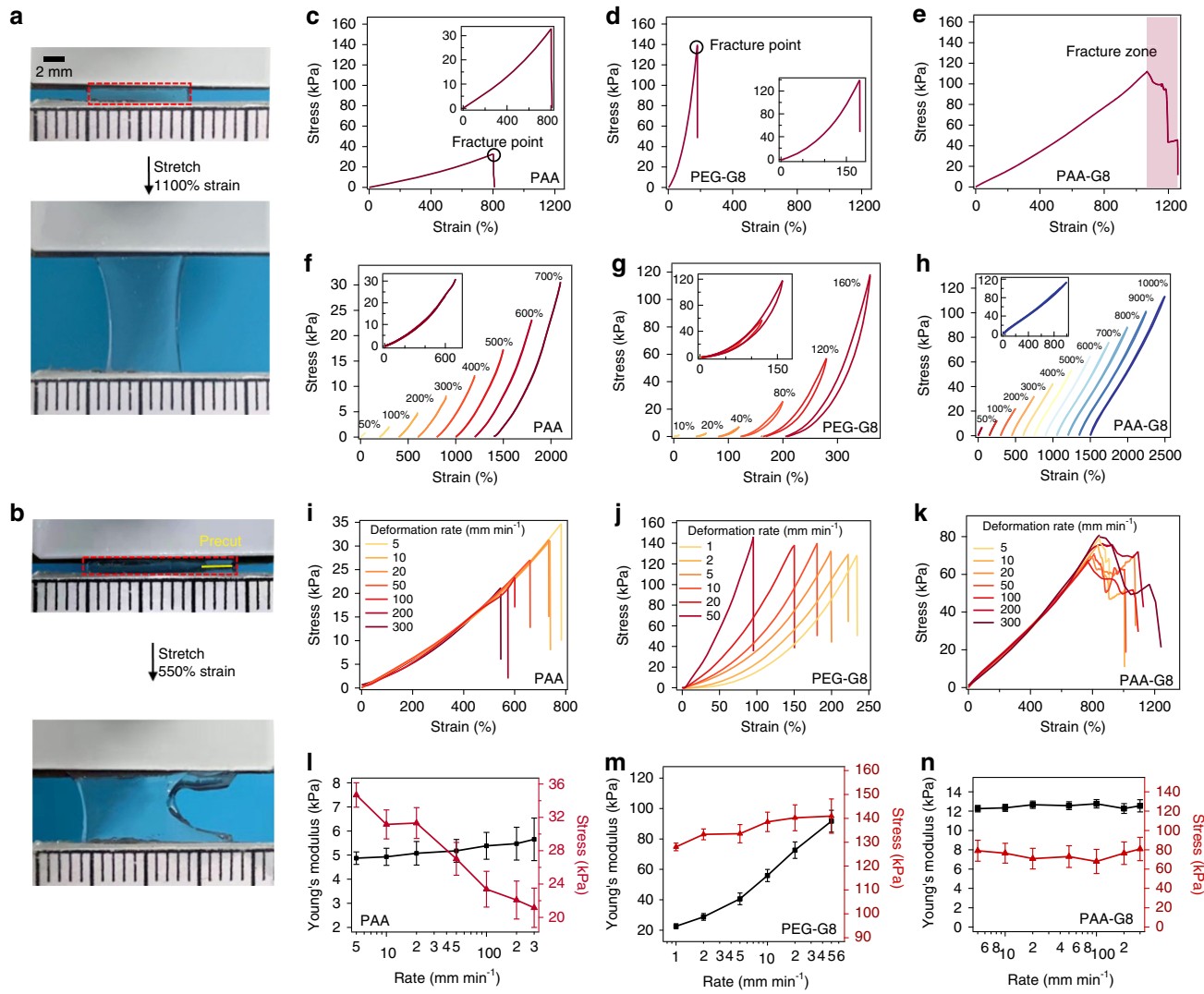

**Fig. 2 The bulk mechanical properties of the designed hydrogels. a** A strip of the undeformed PAA-G8 gel was fixed to two rigid clamps on the top and the bottom. The hydrogel was stretched to 11 times its initial length in a tensile test. **b** A hydrogel with a precut notch (yellow line) was stretched to 5.5 times its initial length without fracture. **c**–**e** The stress–strain curves for the PAA, PEG-G8, and PAA-G8 hydrogels until break. **f**–**h** Representative stretching-relaxation curves for the PAA, PEG-G8, and PAA-G8 hydrogels. The curves are horizontally offset for clarity. The final strains are shown on the curves. Insets show the superposition of the stretching–relaxation curves at different strains. **i**–**k** Stress–strain curves for the PAA, PEG-G8, and PAA-G8 hydrogels at different initial deformation rates. **l**–**n** Fracture stress and Young's modulus of the PAA, PEG-G8, and PAA-G8 hydrogels at different deformation rates. Error bars represent SD.

most GB1 domains were not unfolded even at a strain of 1000% and a stress of 110 kPa. Moreover, we carried out stress-relaxation experiments on the PAA-G8 hydrogels at constant strains. When the PAA-G8 was stretched rapidly to a given strain that was held constant afterwards, there was no clear stress relaxation (Supplementary Fig. 2) even at a strain of 1000%. This further confirmed that GB1 can remain folded in the PAA-G8 hydrogels upon stretching.

Another line of evidence that GB1 domains remained folded in the PAA-G8 hydrogel came from the tensile experiments at varied strain rates (Fig. 2i–k). We did not observe any change of the Young's modulus of the PAA and PAA-G8 hydrogels at the strain rates from 5 to 300 mm min⁻¹, indicating that the strain rates were slower than the speed of the polymer chain uncoiling and no rupture of sacrificial bonds or secondary networks occurred during the stretching process[46,47]. However, in the PEG-G8 hydrogels, the Young's modulus increased greatly with the increase of strain rates, suggesting that GB1 unfolded

irreversibly upon stretching, as the PEG hydrogels without G8 cross-linkers or with unfolded G8 cross-linkers showed constant Young's modulus at varied strain rates (Supplementary Figs. 3 and 4). Moreover, the fracture zone (gradual rupture region, Fig. 2e, k) existed in all PAA-G8 hydrogels, which further suggested that GB1 domains only unfolded locally at the crack sites. The strain-rate-dependent Young's modulus and fracture stress were summarized in Fig. 2l–n. The Young's modulus of the PAA hydrogel was independent of the strain rates but the fracture stress decreased sharply at increasing strain rates. Also, the PAA-G8 hydrogel showed a strain-rate independent Young's modulus and facture stress. This allowed the PAA-G8 hydrogels remaining stretchable at a broad strain rates. It is worth noting that the fracture stress of the PAA-G8 hydrogels was much higher than that of the PAA hydrogels, which can be attributed to the synergistic effects of the presence of folded GB1 domains and their unique network structure.

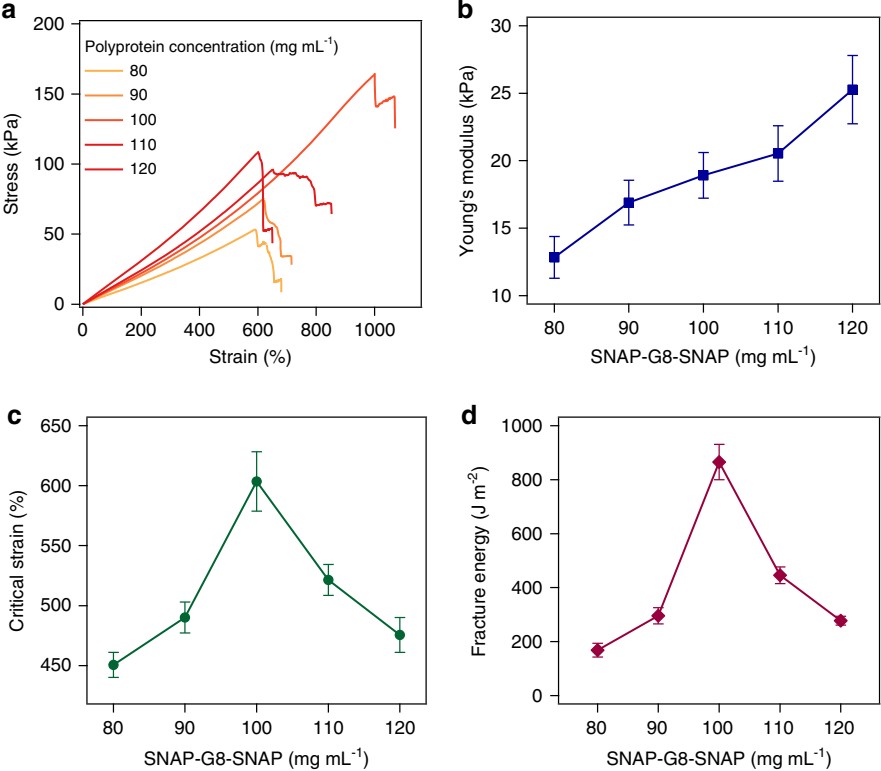

**Fig. 3 Composition greatly affects the mechanical behaviors of the PAA-G8 hydrogel. a, b** Stress–strain curves and the corresponding elastic moduli of the hydrogels of various concentrations of SNAP-G8-SNAP and the same acrylamide (100 mg mL$^{-1}$). The samples were unnotched. **c, d** Critical strain and fracture energy of notched hydrogels of various concentrations of SNAP-G8-SNAP. Error bars represent SD.

The mechanical properties of the PAA-G8 hydrogel were largely dependent on the interplay of the PAA network and the SNAP-G8-SNAP cross-linkers (Fig. 3a). At the fixed PAA concentrations of 100 mg mL$^{-1}$, gradually increasing SNAP-G8-SNAP concentrations could increase the cross-linking density and thus increase the Young's modulus of the hydrogels (Fig. 3b). In contrast, the maximum strain and fracture toughness first increased when the SNAP-G8-SNAP concentration increased from 80 to 100 mg mL$^{-1}$ and then quickly declined upon further increasing the protein concentrations (Fig. 3c, d). We also used notched samples to quantitatively determine the critical strain at which the notch started to extend irreversibly and the fracture toughness which measured the dissipated energy for extending the crack by a unit area (Supplementary Fig. 5). Both critical strain and the fracture toughness reached maxima at a protein concentration of 100 mg mL$^{-1}$. Such a behavior was distinct from that of the PAA hydrogel (Supplementary Fig. 6), which showed a monotonic increase of Young's modulus and decrease of fracture toughness upon the increase of cross-linker concentrations, following the fracture toughness-modulus confliction[48,49]. In the PAA-G8 hydrogel, SNAP-G8-SNAP is not only the cross-linker but also a shock absorber to prevent crack propagation. Therefore, increasing the protein concentrations from 80 to 100 mg mL$^{-1}$ greatly enhanced the critical strain and toughness. However, when further increasing the protein concentrations, PAA was no longer the percolating phase, and some folded proteins were involved in the main network. The Young's modulus of the PAA-G8 hydrogels became strain-rate-dependent, similar to that of the PEG-G8 hydrogels (Supplementary Fig. 7). Both the critical strain and fracture energy dropped down. A similar trend was also observed in the PAA-G8 hydrogels with a fixed protein concentration and varied acrylamide concentrations (Supplementary Fig. 8). The mechanical properties of PEG-G8 hydrogels were

also dependent on the composition, despite that the Young's modulus was consistently higher than that of the PAA-G8 hydrogels at the same SNAP-G8-SNAP concentrations, presumably due to their distinct network structures (Supplementary Fig. 9). Our results highlight the importance of the components and network structure on the mechanical performance of hydrogels. By optimizing the relative concentrations of the percolating polymer networks and the polyprotein cross-linkers, we can obtain highly stretchable hydrogels with low hysteresis and anti-fatigue fracture.

**Visualizing the unfolding of polyprotein cross-linkers in the hydrogel.** To provide direct experimental evidence that GB1 unfolds at the crack-propagation site instead of the entire hydrogel upon stretching, we used an environment-sensitive dye, 1-anilino-naphthalene 8-sulfonate (ANS)[50,51], to spatiotemporally trace the unfolding of GB1 inside the PAA-G8 hydrogels. The fluorescence of ANS became nine times brighter when it bound with the hydrophobic residues exposed upon GB1 unfolding (Supplementary Fig. 10). We immersed the PAA-G8 hydrogel in a PBS buffer containing 100 µM of ANS for 10 min prior to the tensile test. Then, the hydrogel was stretched under the illumination of an ultraviolet lamp (~365 nm) in dark. Figure 4a shows a series of pictures of stretching a PAA-G8 hydrogel with a precut. The entire hydrogel was not fluorescent except for the front of the notch. This clearly indicated that GB1 unfolded at the crack-propagation site instead of the entire hydrogel upon stretching. With the increase of strain, the crack extended gradually and the fluorescent zone became larger. The edge of the notch was also fluorescent indicating that the GB1 proteins at those area were unfolded to resist crack propagation. Since the fluorescence intensity is directly correlated to the amount of GB1 domains unfolded, the fluorescence image also provided an

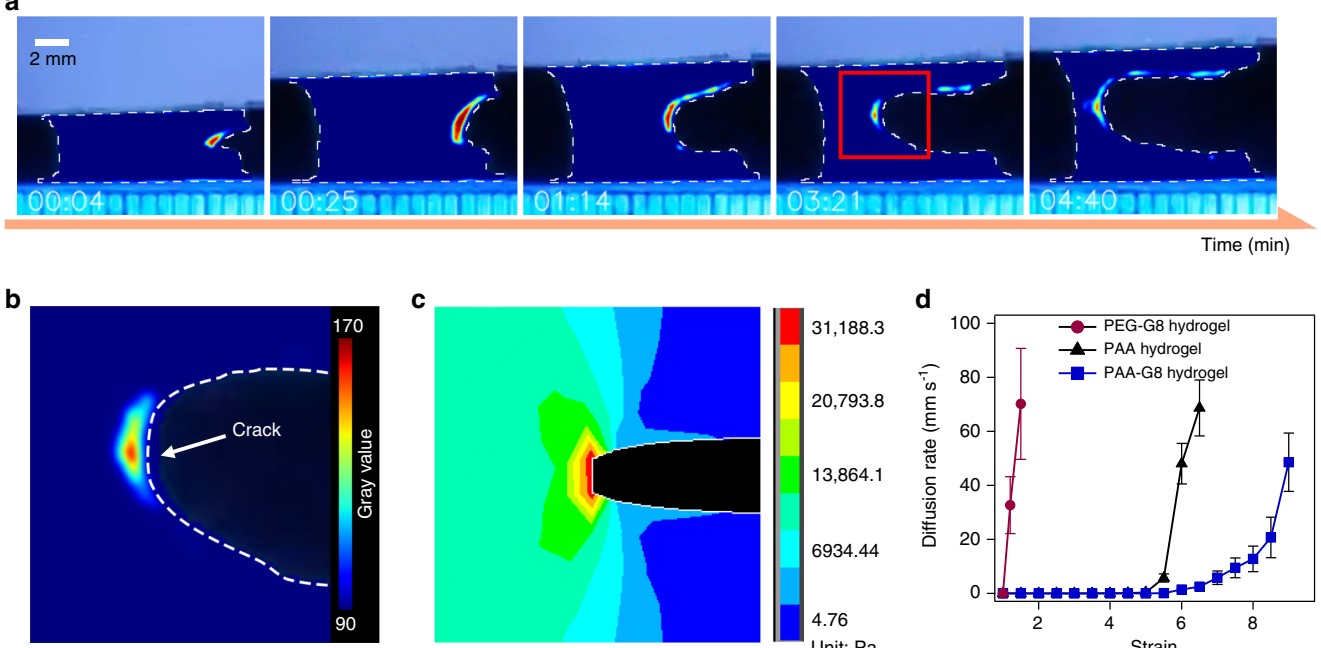

**Fig. 4 Spatiotemporally visualizing the unfolding of polyprotein cross-linkers in the hydrogels. a** Fluorescence intensity images of the precut PAA-G8 hydrogel at different time points along stretching. The hydrogel was loaded with an environment-sensitive dye (ANS) whose fluorescence became brighter in hydrophobic environments. **b** Enlarged photo of the red region in **a**. The white dashed line indicates the crack edge. **c** Stress distribution of a stretched elastic hydrogel simulated using the finite element analysis method. The crack propagation site experienced the highest stress. **d** Crack diffusion rates under different constant strains for the PEG-G8 (purple circle), the PAA (black triangle), and the (blue square) PAA-G8 hydrogels. Error bars represent SD.

opportunity to spatially map the stress propagation within the hydrogel upon stretching (Fig. 4b). Clearly, only GB1 at the crack zone experienced big enough forces to trigger unfolding. We also performed the ANS labeling experiments on intact PAA-G8 hydrogel (Supplementary Fig. 11 and Supplementary Movie 2). The results show that GB1 largely remained folded before the hydrogel reached the fracture limit. A few fluorescent spots at the edge of the hydrogel can be seen during stretching, which may indicate stress-concentration at those area due to the presence of defects. To quantitatively understand the stress-concentration at the crack tip in the notched hydrogel, we used finite element analysis to simulate the stress distribution (Fig. 4c). Our simulation result demonstrated that the stress at the crack propagation site was ~31 kPa, significantly higher than the rest part (~14 kPa). In contrast, when dying with ANS, the PEG-G8 hydrogel quickly became fluorescent upon stretching, as GB1 was involved in the percolating phase and the stress quickly reached the threshold for GB1 unfolding (Supplementary Figs. 12, 13 and Supplementary Movies 3, 4).

By recording the crack propagation of the PAA-G8 hydrogels with a precut notch under different constant strains, we determined the crack diffusion rate (Fig. 4d). The crack started to diffuse at much higher strains in the PAA-G8 hydrogel than the PAA and PEG-G8 hydrogels. Moreover, the crack diffusion rate increased very slowly with the increase of strain rates even when the strains were beyond the fracture threshold (i.e., at strains from 6 to 8). In contrast, the crack diffusion rates of the PAA and PEG-G8 hydrogels increased sharply after reaching the threshold. This further suggested that the PAA-G8 hydrogels can significantly prevent fatigue fracture.

Furthermore, we have performed the stretching experiments on the PAA-G8 hydrogels with varied G8 concentrations in the presence of ANS (Supplementary Figs. 14, 15 and Supplementary Movies 5, 6). Our results clearly showed that at the G8 concentration of 120 mg mL$^{-1}$, PAA was no longer the percolating phase and the hydrogel was lighted up upon stretching due to the unfolding of GB1. In contrast, at a lower G8 concentration of 80 mg mL$^{-1}$, PAA remained as the percolating phase and the whole hydrogel, except for the crack area, kept dim. These results further suggest that having PAA as the percolating phase is critical for achieving low hysteresis, high stretchability, and high fatigue resistance.

**Characterization of fatigue fracture of hydrogels.** Next, we measured the anti-fatigue fracture properties of the hydrogels upon cyclic load/unload following the test procedures reported by Zhang et al.[52]. For all three kind of hydrogels, the strain was cycled between 1 and 5. The strain rate of the test was fixed to 5 s$^{-1}$. To record the crack extension, we recorded the pictures of the hydrogel every 100 cycles. Figure 5a shows the photos of the hydrogel in the first and the 5000th cycle. Even after 5000 cycles, we did not observe any measurable crack propagation (Fig. 5b). All stress–strain curves in the load/unload cycles were superimposable without obvious hysteresis (Fig. 5c). However, for the PAA hydrogel, the crack extended rapidly and ran through the hydrogel within five cycles. For the PEG-G8 hydrogel with a precut, the crack ran through quickly upon stretching, and the sample did not survive in the cycling test. Finally, we applied cyclic stretch on the precut sample of the three kinds of hydrogels to different strains and recorded the extension of crack cycle by cycle to calculate the fatigue thresholds of these hydrogels (Supplementary Fig. 16). Based on these data, the extensions of crack per cycle, d$c$/d$N$, as a function of the energy release rate, $G$, are plotted in Fig. 5d. The results show that the fatigue threshold of PAA-G8 hydrogel is about 126 J m$^{-2}$, which is much higher than that of the PAA (7.5 J m$^{-2}$) and the PEG-G8 hydrogels (14.2 J m$^{-2}$). Moreover, the fatigue threshold of the PAA-G8 hydrogel is comparable to that of the crystallinity toughened hydrogels of the same dry weight[16] and the double-network hydrogels containing weak

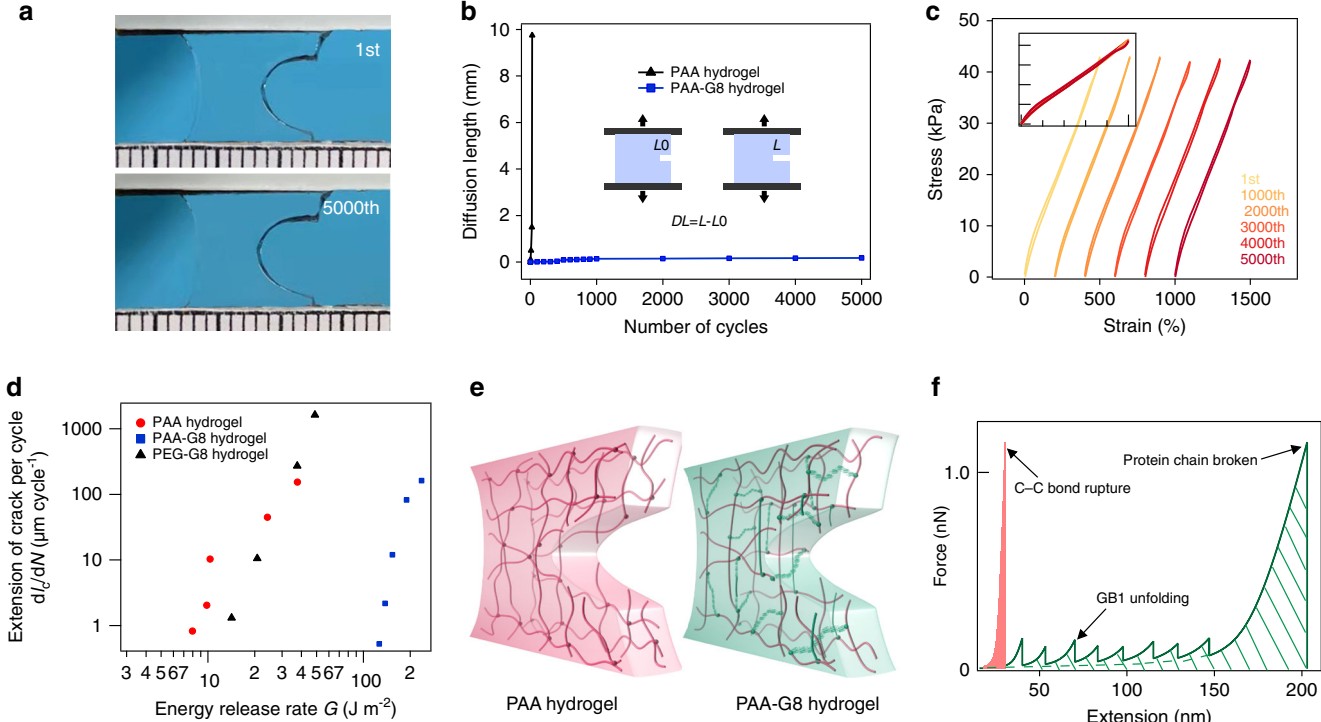

**Fig. 5 Characterization of the anti-fatigue fracture properties of the hydrogels. a** Photos of the 1st and 5000th cycles of the PAA-G8 hydrogel with a precut at 5 strain. **b** Fatigue crack diffusion of the PAA (black triangle) and the PAA-G8 hydrogel (blue square) over different number of loading cycles. **c** Stress–strain curves of the PAA-G8 hydrogel maintained the same shape in thousands of loading cycles. **d** Extension of crack per cycle as a function of the energy release rate. **e** Schematic illustration of the unfolding of polyprotein cross-linkers to prevent crack propagation in the PAA-G8 hydrogel but not in the PAA hydrogel. **f** Force-extension curves of unstructured PAA and the G8 protein. PAA can only be extended to a short extension before the rupture of the covalent bond of the polymer. The mechanical energy required to break PAA is small (shade in light orange). Stretching of G8 leads to the sequential unfolding of GB1 domains, giving rise to saw-tooth-like force-extension pattern. The polyprotein behaves like a shock-absorber and dissipates considerable energy before rupture (shade with oblique lines).

sacrificial bonds[45]. Yet, the PAA-G8 hydrogel did not show obvious mechanical hysteresis in the cyclic stretch experiments.

## Discussion

Inspired by the structure of natural load-bearing tissues, tandem-repeat proteins have been recently used as the building blocks of many synthetic hydrogels[24,25]. In those hydrogels, the folded tandem-repeat proteins and the other unstructured proteins or synthetic polymers were combined together to form the percolating phase. The mechanical response of the folded proteins is coupled with the mechanical deformation of the hydrogel network. The folded protein domains experienced considerable tensile forces and unfolded upon stretching. They cannot completely fold back during relaxation, leading to obvious hysteresis. Such a hysteresis loop was obviously beneficial for damping mechanical stress and increasing toughness. However, for the applications as sensors and actuators, low hysteresis and anti-fatigue properties are more preferred. Here we show that by hydrogel structure engineering, we are able to decouple the network elasticity with local mechanical response of the polyprotein cross-linkers. By imbedding the folded polyprotein cross-linker in a percolating random coiled PAA network, the hydrogel can be stretched up to 1100% strain with a hysteresis of <5%. Moreover, the hydrogel show outstanding anti-fatigue properties with a fatigue fracture threshold of 126 J m⁻². 

$126 \, \text{J m}^{-2}$.

Our results highlight the importance of hydrogel network structure on the mechanical behaviors of protein-based hydrogels. Since the polyprotein cross-linkers are not involved in the percolating phase, they do not experience sufficiently high forces

to unfold until the hydrogels approach the fracture limit. This is evidenced by (1) the low hysteresis of the stress–strain curves, (2) the strain-rate independent elastic properties, and (3) the fluorescent labeling experiments. Therefore, the polyprotein cross-linkers can endow considerable fracture toughness to the hydrogels without introducing hysteresis. Such a design breaks the hysteresis-toughness correlation that is usually reported in stretchable and tough hydrogels.

On the other hand, the polyprotein cross-linkers also make the hydrogel more stretchable and defect-insensitive by reshaping the network structure. In real hydrogel networks, short loops formed by polymerization of the bi-functional cross-linkers to the same polymer chain often have the advert effects to the stretchability and elasticity of the hydrogels. However, the end-to-end distance of the SNAP-G8-SNAP cross-linkers is ~30 nm, allowing the two termini well separated. This greatly minimizes the probability of forming short loops in the network. Indeed, the PAA-G8 hydrogels exhibit higher Young's modulus, break strain, and fracture toughness than the PAA hydrogel. Similarly, in the previous study of actin network[53], Bausch and coworkers also revealed that the critical strain and fracture stress can be greatly increased by increasing the number of the spacing units within the cross-linkers.

The most significant impact of the polyprotein cross-linker lies in the increase of the fatigue threshold of the hydrogel. The stress–strain curves showed clear signature of stepwise fracture, which was rarely observed in other types of anti-fatigue hydrogels. Theoretically, the fatigue threshold of hydrogels can be estimated using the Lake–Thomas model, which assumes that crack grows by breaking a single layer of polymer chains at the

crack zone[54]. Suo and coworkers have provided a quantitative model, Eq. (1), to estimate the fatigue threshold of hydrogels[45,55]

$$\Gamma_0 = \phi_{PAA}^{2/3} bUln^{1/2}, \tag{1}$$

where $\phi_{PAA}$ is the volume fraction of PAA network in the hydrogels, $b$ is the number of bonds in the polymer main chain per unit volume of the dry polymer, $U$ is the energy of the C–C bond, $l$ and $n$ are the length of the monomer and the number of monomer in a PAA chain, respectively. Based on this model, the PAA hydrogel has a fatigue threshold of 5.1 J m$^{-2}$ (see Supplementary Information). Because GB1 domains are already unfolded, the fatigue threshold of the PEG-G8 hydrogels is estimated to be 10 J m$^{-2}$[56]. However, in the PAA-G8 hydrogel, the unfolding of G8 prior to the fracture of the PAA chain should be considered. As shown in Fig. 5e, f, the effects of the polyprotein cross-linkers are twofolds. First, it dissipates the mechanical energy by unfolding protein domains sequentially. Second, it increases the effective bond numbers per unit volume of the dry polymer. Both effects can lead to considerable increase of the fatigue threshold. The polyproteins are randomly distributed in the fracture zone and only the cross-linkers perpendicular to the crack growth direction are subjected to stretching forces and unfold (Fig. 5e). The cross-linkers at the parallel positions experience lower strains and do not unfold. By considering these effects, the fracture threshold is calculated to be 138 J m$^{-2}$, which is close to the experimentally determined value (126 J m$^{-2}$) (see Supplementary Fig. 17 and Supplementary Information for calculation details). It is worth mentioning that in the original Lake–Thomas model, except for chain scission, other energy dissipation (e.g., viscoelasticity, poroelasticity, and protein unfolding) in real soft materials is not considered. The way we estimated the energy dissipation based on single molecule force spectroscopy data may have certain systematic errors due to the assumption of the strain rates during crack propagation and the complexity of the network structures[57]. Some protein domains may remain folded before the breakage of the cross-linker, if the local strain rate is too fast. The model should be further improved in the future to provide quantitative prediction of the fracture threshold. Nonetheless, the calculation further suggests that the polyprotein cross-linkers contribute greatly to the fatigue threshold but little to the hysteresis. This is distinct from the behaviors of tough hydrogels that have been widely explored recently[45].

Besides these advances in hydrogel design, we also provide an experimental tool to track forced protein unfolding in hydrogels in real time. Using a fluorescent dye, ANS, to specifically bind with the hydrophobic residues of unfolded GB1, we monitored the unfolding of GB1 within the PAA-G8 hydrogels with high spatiotemporal resolution. Upon stretching, we clearly observed that the fluorescence intensity only considerably increased at the tip of the crack and remained dim on the rest part of the hydrogels. Elemental mechanical analysis revealed that the position of the GB1 unfolding correlated well with the location in the hydrogel that experiencing high mechanical stress. Previously, Creton and coworkers have elegantly demonstrated that the mechanical stress within a soft material can be probed using mechano-sensitive fluorophores[1]. Due to the fast binding of ANS to the hydrophobic residues of unfolded proteins, we propose that this method can be also used to probe the mechanical forces within various hydrogel materials. Especially, the unfolding forces of proteins can vary markedly, allowing the protein-based force sensor to function over a broad force range. Note that, the ANS-based protein unfolding sensor shows a fluorescence turn-on feature with low background fluorescence.

In summary, we have demonstrated a principle for engineering highly stretchable, low-hysteresis, and anti-fatigue fracture hydrogels. Using flexible polymers as the percolating phase and the mechanically strong polyproteins as the cross-linker, we can specifically increase the fatigue threshold without affecting the resilience of the hydrogels. It is worth mentioning that the G8 cross-linker is stable in pure water and tolerable to dehydration. The hydrogel can be dehydrated and rehydrated in water without causing significant changes to the mechanical properties (Supplementary Figs. 18 and 19). We anticipate that these hydrogels can find broad applications in soft robotics, flexible sensors, and smart wearable devices, where the materials are routinely subjected to multiple load/unload cycles.

## Methods

**Protein engineering**. The gene encoding protein SNAP-(GB1)$_8$-SNAP were constructed in pQE80L vectors using standard molecular biology techniques. The proteins were expressed in *E.coli* (BL21) and purified by Co$^{2+}$-affinity chromatography. The proteins were dialyzed into deionized water and lyophilized before use.

**Synthesis of BS**. The details of the synthetic procedures of BS and characterizations (Supplementary Figs. 20–23) were described in Supplementary Information.

**Hydrogel preparation and mechanical test**. To prepare the PAA-G8 and the PEG-G8 hydrogels, BS was covalently conjugated to SNAP-G8-SNAP by mixing them with a molar ratio of 2:1 in a shaker at 4 °C overnight. Then the BS-linked SNAP-G8-SNAP was mixed with acrylamide or 4-armed-PEG-SH at desired concentrations. Next, LAP (0.05%) was added into the two kinds of mixtures and transferred to a custom-made transparent glass mold with a thickness of 1 mm. The polymerization was proceeded under UV illumination for 1 h at room temperature. Subsequently, the formed hydrogels were taken out from the mold and soaked in PBS at 4 °C for 24 h to reach the equilibrium-swollen state. The swelling ratios were ~1.6 for all hydrogels. Tensile tests were performed using an Instron-5944 tensometer with a 10-N static load cell at room temperature.

**Finite element analysis simulation**. We use a commercial finite-element software, ANSYS, to simulate the stress states. Two parameters—Young's modulus $\mu$ and Poisson's ratio $\nu$ were identified as $12 \times 10^3$ Pa and 0.5, respectively. We used the quad element in ANSYS and modeled the hydrogel as an isotropic elastic material. The geometry was set as a thin sheet with $L = 10$ mm, $H = 2$ mm according to the geometry used in real experiments. A cut existed at the middle of the right side with a length of 1 mm. The sample was stretched to four times of its initial length, and the stress distribution was calculated.

## Data availability
The data that support the findings of this study are available from the corresponding author upon request. The source data underlying Figs. 2–5 are provided as Source Data file.

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

## Acknowledgements

This research is supported mainly by the National Natural Science Foundation of China (Grants Nos. 11804147, 11974174, 11934008, 21774057, 11804148, and 11674153), Youth Program of Natural Science Foundation of Jiangsu Province (Grant Nos. BK20180335 and BK20180320), Fundamental Research Funds for the Central Universities (Grant Nos. 020414380154, 020414380080, and 020414380148), and Project supported by State Key Laboratory of Precision Measurement Technology and Instruments (Tianjin University).

## Author contributions

Y.C., H.L., and W.W. conceived the project and designed the experiments. H.L., L.D., J.Z., H.C., Y.Z., and B.X. performed the hydrogel experiment and analyzed the data. Y.L. synthesized the chemicals. J.W. performed the molecular biology experiments. Q.F. and B.C. designed and performed theoretical modeling. Y.C., W.W., and M.Q. supervised the project. H.L. and Y.C. wrote the paper with contributions from all authors.

## Competing interests

The authors declare no competing interests.
