## [Peer Review File · Nature Communications]

Reviewers' comments:

Reviewer #1 (Remarks to the Author):

In this work, the authors provide a route to decouple network elasticity and local mechanical response in synthetic hydrogels. They developed a hydrogel using tandem-repeat proteins as the cross-linkers and random coiled polymers as the percolating network. Such a design allows the polyprotein crosslinkers only to experience considerable forces at the fracture zone and unfold to prevent crack propagation, which decouples the hysteresis-toughness correlation and create highly stretchability and tough hydrogels with negligible hysteresis. Moreover, the hydrogels also show a high fatigue performance. Although the mechanism is not fully clear yet, this is really an excellent work and deserves publication in Nature Communications.

Specific comments:

While the experimental evidence that polyprotein crosslinkers in PAA-G8 only to experience considerable forces at the fracture zone and unfold to prevent crack propagation is sound, the reviewer cannot understand the explanation of why the polyprotein crosslinkers do not unfold at tensile deformation even at large strains. At the crack tip, usually the sample also experience tensile deformation. Moreover, the structures illustrated in Figure 1b are used to explain why PEG-G8 results in unfolding while the PAA-G8 does not by tensile deformation. This explanation of different responses to tensile deformation is also not understandable. No matter at as the crosslinker or as part of the polymer chain, the polyprotein should experience the same tensile force at large deformation, especially when the chain is stretched near its extremity.

Reviewer #2 (Remarks to the Author):

This manuscript described the fabrication of an anti-tearing and anti-fatigue fracture hydrogel with low hysteresis. The basic mechanism is the unfolding of proteins during crack propagation to dissipate energy and thus prevent the crack growth. Unlike previous examples, the authors smartly used the energy dissipating proteins only as cross-linker, rather than incorporate them in the whole network to function also as percolating chains. This design could maintain the proteins folded during stretch until reaching the critical point. Thus the hysteresis was lowered by skipping unfolding-refolding process, and the stepwise unfolding of protein provided better anti-tearing properties along with anti-fatigue fracture properties. The general idea and performance of the hydrogels were good, but I do not think it is up to the standard of Nature Communication. The following are to be explained, clarified or corrected, and some essential data are to be provided.

1. "The Young's modulus of the PAA hydrogel was independent of the strain rates but the fracture stress decreased sharply at increasing strain rates because the PAA hydrogel lacked a mechanism to prevent crack propagation." The decrease in fracture stress with strain rates is not because of crack propagation. It's due to time scale of polymer chain uncoiling. "For the PEG-G8 hydrogel, the Young's modulus increased considerably with the increase of strain rates, presumably due to the unfolding of GB1." I don't think this is correct. The process can be very complicated, better not to discuss the reason without direct evidence.

2. In the paragraph before figure 3, the authors said "the hydrogel behaved like the PEG-G8 hydrogel". The authors should explain "behaved" in which way. Low hysteresis? But it's discussing fracture toughness here. Fracture toughness? But the critical strain and fracture energy of PEG-G8 were not provided anywhere in the manuscript. And the critical strain was too low of PEG-G8, it's hard to say any of the PAA-G8 behaved like PEG-G8 in this way. Moreover, the critical strain and fracture energy of PEG-G8 should also be provided.

3. "It is worth noting that the fracture stress of the PAA-G8 hydrogels was much higher than that of the PAA and PEG-G8 hydrogels". But in figure 2, the fracture stresses were about 30 for PAA (c&l), 140 for PEG-G8 (d&m), 100 (e) or 80 (n) for PAA-G8. PEG-G8 is the highest.

4. The authors used "toughness" and "fracture toughness" in the manuscript, they could have different meaning. Be sure they were clarified. For example, "In contrast, the maximum strain and toughness first increased...". This description is more likely for intact samples, especially followed by "We also used notched samples...". But fig. 3c and d were for notched samples. This part is very misleading and confusing. And also for other parts, "fracture toughness" should be used instead of "toughness" if discussing notched samples and anti-tearing properties.
5. "Moreover, the diffusion rate increased very slowly with the increase of the strain. Contractively, the crack diffusion rate increased sharply after reaching the threshold. This further suggested that the PAA-G8 hydrogels can significantly prevent fatigue fracture." The description here is also confusing. How could these phenomena suggest "the PAA-G8 hydrogels can significantly prevent fatigue fracture"? Did the author want to compare the "slow increase of diffusion rate of PAA-G8 from strain 6 to 8" with "sharply increased diffusion rate of PAA gel after the threshold"? Please state clearly.
6. In figure 4, the unfolding could be easily monitored through fluorescence. A similar experiment stretching intact PAA-G8 gel with ANS would provide more convincing evidence and more insight about the process. And with this consideration, the "(3) the fluorescent labeling experiments" "until the hydrogels approach the fracture limit" mentioned in discussion part was not provided.
7. The stretching images of notched PEG-G8 gel with ANS in Figure S7 does not show clear precut as for PAA-G8. More representative and persuasive images and video should be attached. Also, stretch results of an intact PAA-G8 gel with ANS are necessary.
8. "Based on these data, the extensions of crack per cycle, dc/dN , as a function of the energy release rate, G , are plotted in Fig. 5d", Please provide in SI how the calculation was carried out according to the experimental results.
9. In the "Characterization of fatigue fracture of hydrogels" part, what's the cycled strain for PAA and PEG-G8? The one (5.5) for PAA-G8 already exceeded the critical strain of these two gels. And "Even after 5000 cycles, we did not observe any measurable crack propagation". But in figure S8c, obvious propagation was demonstrated with 5.5 strain, any explanation?
10. In discussion, "This is reasonable because some protein domains may be already unfolded prior to reaching the fracture limit". But in the whole manuscript, "Since the polyprotein cross-linker does not form the percolating phase, they do not experience sufficiently high forces to unfold until the hydrogels approach the fracture limit" is a key point, and suggested by experiment results. Is it suitable to use it as a reason (without evidence and in contradiction with previous arguments) for the difference between theoretical and experimental value? And 272 is not just "a bit" higher than 126. Please provide more reasonable explanation.
11. In discussion "the calculation further confirm that the polyprotein cross-linkers contribute greatly to the fatigue threshold but little to the toughness". First, how did the calculation "confirm" it? The analysis is missing. Second, PAA cross-linked by Bis showed both much lower toughness of intact samples (strain energy density, not calculated but obvious from the area under the curve in fig 2c and e) and fracture toughness (fracture energy, ~ 35 compared with ~ 800) than PAA-G8. The most difference is the polyprotein cross-linkers. How could it contribute "little to the toughness"?

Some points about the writing:

1. The introduction is quite lengthy and much irrelevant information can be omitted or shortened, such as the evolution of tough hydrogels and examples of folded proteins in natural biological tissues. However, the authors should make it clear why achieving low hysteresis, high stretchability, and high fatigue resistance, concurrently in a single hydrogel is so difficult from a fundamental point of view, and why your design is smart and unique in overcoming this difficulty. Although there are some explanations on the latter in the last paragraph of introduction, it focuses on crack propagation with too much detail and does not cover all three properties.
2. In the introduction, "Therefore, these folded protein domains can efficiently prevent fracture propagation through unfolding without causing additional hysteresis". This sentence is very confusing. The protein is folded or unfolded? Why unfolding does not cause hysteresis? Although reading the rest of the article can give a clear picture of this work, these summary sentences are very important and should be rewritten.

Reviewer #3 (Remarks to the Author):

Comments

The manuscript written by Hai Lei and Liang Dong, et al. describes a fascinating design concept of a hydrogel that possesses an ability to decouple the hysteresis-toughness correlation. To achieve this, the authors utilize tandem-repeat proteins (called G8) as the functional cross-linker which works as a shock-absorbable architecture. When forces acting on the polymer network reach a critical value, unfolded G8 units fold to prevent mechanical fatigue and dissipate energy. This concept has already been reported as mentioned in the manuscript (ref 24, 25), however, the authors newly find that a specific network structure enables the ability to decouple the network elasticity with local mechanical response of the G8 cross-linker. Specifically, the folded G8 cross-linker is embedded in a percolated random coiled polyacrylamide network (called PAA-G8 hydrogel) to separate G8 units from the percolated phase. This unique structure prevents G8 unit folding until the hydrogels approach the fracture limit, resulting in excellent mechanical properties such as high fracture strain (~1100%) with low hysteresis (>5%). To verify the relationships between the network structure and mechanical properties, the authors fabricate another type of hydrogel which contain G8 units in the percolated phase (called PEG-G8 hydrogel). By measuring the mechanical properties and comparing the results of PAA-G8, PEG-G8 and neat PAA hydrogels, the effect of network structure is successfully revealed. Furthermore, the authors establish the method to visualize the protein unfolding during tensile testing by utilizing an environmentally sensitive dye (called ANS). These results agree well with the simulations performed by the finite element analysis method. Finally, fatigue fracture of hydrogels are systematically characterized by cyclic loading/unloading tests. Among them, the PAA-G8 hydrogel exhibits outstanding anti-fatigue properties even after 5000 cycles with a fatigue fracture threshold of 126 Jm^{-2} . By comparing this value and the theoretically calculated one, the experimental results are a bit lower but reasonable, taking protein folding stability into consideration.

I believe that the main research findings of this paper will be important towards understanding network structures containing tandem-repeat units and designing highly stretchable, low hysteresis and anti-fatigue hydrogels. Moreover, the subject matter of this work is laudable and of interest to various science communities such as soft robotics, flexible sensors and smart wearable devices.

Therefore, I would recommend this paper for publication in Nature Communications; however, several points as indicated below should be addressed by the authors to improve the quality of the article.

1. In the section [Mechanical characterization of the PAA-G8 hydrogels], the authors systematically change the protein cross-linker concentration. When the concentration exceeds a certain point, the authors mentioned that PAA is no longer the percolated phase (some folded proteins were involved in the main network). Could you verify this by using a visual method as the authors established?

2. In Fig 4, we can clearly observe florescence indicating protein folding around the crack-propagation site. If the author do the same experiment without precut, how does the fluorescent intensity change? You mentioned that G8 units in PAA-G8 gel do not experience sufficiently high force to unfold until the hydrogel approaches the fracture limit. Could you verify this by your method? Also, how about the intensity change in the case of PEG-G8?

3. To stabilize the protein unit, the hydrogels were swollen in PBS. I'm curious about the environmental stability of G8 cross-linker. Does it work in pure water? Does it work even after drying then reswelling? If the G8 cross-linker is environmentally durable, it might be good for applications such as wearable devices and soft robotics.

Minor points:

In Fig.2(c-e),(f-h),(i-k),(l-n), it is little bit hard to tell which date represents which gel. Could you put a label or subheading in the figure?

Point by point response

Reviewer #1:

In this work, the authors provide a route to decouple network elasticity and local mechanical response in synthetic hydrogels. They developed a hydrogel using tandem-repeat proteins as the cross-linkers and random coiled polymers as the percolating network. Such a design allows the polyprotein cross-linkers only to experience considerable forces at the fracture zone and unfold to prevent crack propagation, which decouples the hysteresis-toughness correlation and create highly stretchability and tough hydrogels with negligible hysteresis. Moreover, the hydrogels also show a high fatigue performance. Although the mechanism is not fully clear yet, this is really an excellent work and deserves publication in Nature Communications.

Response:

We thank the reviewer for his/her positive comments.

Comment 1:

While the experimental evidence that polyprotein cross-linkers in PAA-G8 only to experience considerable forces at the fracture zone and unfold to prevent crack propagation is sound, the reviewer cannot understand the explanation of why the polyprotein cross-linkers do not unfold at tensile deformation even at large strains. At the crack tip, usually the sample also experience tensile deformation.

Response:

We thank the reviewer for this comment. We agree with the reviewer that we did not explain the mechanism clearly in the original manuscript. In our understanding, the macroscopic deformation of the PAA-G8 hydrogel is mainly contributed by the extension of the percolating polyacrylamide phase but not the G8 cross-linkers, as the G8 cross-linkers are bypassed. The major cross-links that bear considerable forces are the entanglement points of the PAA chains. The forces only propagate to the G8 cross-linkers when the PAA chains are considerably tightened. Even at large strains, the extension of the G8 cross-linkers is small. As the reviewer pointed out, at the crack tip, the local tensile deformation is larger than that in the whole hydrogel. The polyacrylamide chains are almost fully tightened and forces start to propagate to the G8 cross-linkers, causing their unfolding. Please refer to the new Fig. 1b for the schematic illustration of this mechanism. In short, due to the unique network structure of the PAA-G8 hydrogels, the deformation of the network can only lead to the sharp increase of stretching forces on the G8 cross-linkers at extremely large strain beyond the fracture strain of the hydrogel. However, this strain limit can be met at the crack tip due to the stress concentration effect. We now have added this explanation in the revised manuscript.

Revision:

Fig. 1b. Illustration of the network structures of the PAA, PEG-G8, and PAA-G8 hydrogels before and after stretching to the critical crack propagation point. The green rods represent folded polyproteins and green lines represent unfolded proteins. In the PAA and PEG-G8 hydrogels, the fatigue threshold is determined by the energy required to fracture a single layer of polymer chains per unit area. In the PAA-G8 hydrogel, the fatigue crack propagation requires both the fracture of polymer chains and the unfolding of polyproteins at the single layer of the crack front. (On page 4 of the revised manuscript)

“However, in the PAA-G8 hydrogels, the macroscopic deformation of the hydrogels is mainly contributed by the extension of the percolating polyacrylamide phase but not the G8 cross-linkers, as the G8 cross-linkers are bypassed (Fig. 1b). The major cross-links that bear considerable forces are the entanglement points of the PAA chains. The forces only propagate to the G8 cross-linkers when the PAA chains are considerably tightened. Even at large strains, the extension of the G8 cross-linkers is small. The deformation of the network can only lead to the sharp increase of stretching forces on the G8 cross-linkers at extremely large strain beyond the fracture strain of the hydrogels. However, this strain limit can be met at the crack tip due to the stress concentration effect. The unfolding of GB1 then greatly prevents the crack propagation.” (On page 3 of the revised manuscript, last paragraph)

Comment 2:

The structures illustrated in Figure 1b are used to explain why PEG-G8 results in unfolding while the PAA-G8 does not by tensile deformation. This explanation of different responses to tensile deformation is also not understandable. No matter at as the cross-linker or as part of the polymer chain, the polyprotein should experience the same tensile force at large deformation, especially when the chain is stretched near its extremity.

Response:

We thank the reviewer for this comment. We apologize for not illustrating the structures of the two hydrogels clearly in Figure 1b. The reviewer is correct that the polyprotein should experience the same tensile force when the chain is stretched near its extremity. However, the PEG-G8 hydrogel and the PAA-G8 hydrogel have different network structures, leading to distinct forces on the polyprotein at the same macroscopic strains. In the PAA-G8 hydrogels, the G8 cross-linkers are bypassed in the percolating PAA network. In contrast, in the PEG-G8 hydrogels, the G8 polyproteins are integrated in the PEG network. We have redrawn the figures to explain the difference.

We'd also like to share some our on-going theoretical work in predicting the mechanical response of the PAA-G8 and PEG-G8 hydrogels based on their network structures. The theoretical model (representative volume element), predicted stress-strain curves, and the unfolded fractions of GB1 domains in the two hydrogels are shown in Fig. R1. The theoretical analysis indicates that most of the GB1 domains in the PAA-G8 hydrogel remain folded upon stretching and the stress-strain curves in the stretching-relaxation cycles are superimposable. Even at a strain of 1100%, only 0.026% of GB1 domains are unfolded. However, in the PEG-G8 hydrogel, even at the initial free swelling state, 17.9% of the GB1 domains are unfolded. Upon loading, the hysteresis of the stress-strain curves increases with the increase of strain. At a strain of 160%, the fraction of unfolded GB1 domains increase to ~38%. The theoretical model can semi-quantitatively recapture the mechanical properties of the two hydrogels and reveals that their distinct mechanical response stems from their disparate network structures. At current stage, we are still working on the prediction of the fracture energy using this model. The whole theoretical model will be published elsewhere in the future.

Fig. R1. Theoretical calculations of the mechanical response of the PAA-G8 and PEG-G8 hydrogels. **a, b**, Representative volume element of a cube for gel elasticity. **c, d**, Prediction of stress–strain curves for the PAA-G8 (**c**) and PEG-G8 (**d**) hydrogels upon uniaxial loading and unloading cycles. Only the PEG-G8 hydrogel is shown to dissipate a large amount of energy within loading cycles. **e, f**, Prediction of the unfolded fractions of GB1 domains within the PAA-G8 (**e**) and PEG-G8 (**f**) hydrogels upon uniaxial loading.

Revision:

See new Fig. 1b as attached to the response to the previous comment. (On page 4 of the revised manuscript)

Reviewer #2 (Remarks to the Author):

This manuscript described the fabrication of an anti-tearing and anti-fatigue fracture hydrogel with low hysteresis. The basic mechanism is the unfolding of proteins during crack propagation to dissipate energy and thus prevent the crack growth. Unlike previous examples, the authors smartly used the energy dissipating proteins only as cross-linker, rather than incorporate them in the whole network to function also as percolating chains. This design could maintain the proteins folded during stretch until reaching the critical point. Thus the hysteresis was lowered by skipping unfolding-refolding process, and the stepwise unfolding of protein provided better anti-tearing properties along with anti-fatigue fracture properties. The general idea and performance of the

hydrogels were good, but I do not think it is up to the standard of Nature Communication. The following are to be explained, clarified or corrected, and some essential data are to be provided.

Response:

We thank the reviewer for his/her critical comments and valuable suggestions. Following the reviewer's advice, we have performed additional experiments and revised the manuscript substantially. We have also improved the writing according to the reviewer's suggestions. We hope that the reviewer will find the revised manuscript can meet the standard of Nature Communications.

Comment 1:

"The Young's modulus of the PAA hydrogel was independent of the strain rates but the fracture stress decreased sharply at increasing strain rates because the PAA hydrogel lacked a mechanism to prevent crack propagation." The decrease in fracture stress with strain rates is not because of crack propagation. It's due to time scale of polymer chain uncoiling. "For the PEG-G8 hydrogel, the Young's modulus increased considerably with the increase of strain rates, presumably due to the unfolding of GB1." I don't think this is correct. The process can be very complicated, better not to discuss the reason without direct evidence.

Response:

We thank the reviewer for pointing out the correct mechanism. We totally agree with the reviewer that the decrease in fracture stress with strain rates for the PAA hydrogel is because that the strain rates become comparable to or even faster than the speed of polymer chain uncoiling. We have removed this discussion in the revised manuscript according to the reviewer's suggestion.

To prove that the increase of Young's modulus of the PEG-G8 hydrogels with the strain rates is due to the unfolding of GB1, we performed a control experiment using the PEG hydrogel cross-linked by a short covalent cross-linker (dithiothreitol, DTT) without the polyprotein GB1 domains (named as the PEG-DTT hydrogel). The Young's modulus of this hydrogel was independent on the strain rates (Fig. S2). We have also measured the mechanical response of the PEG-G8 hydrogel in the presence of a chemical denaturant, urea, which can completely unfold GB1. Under the denaturing conditions (GB1 remained unfolded), the Young's modulus of the PEG-G8 hydrogel was significantly reduced and became independent on the strain rates (Fig. S3). Taken together, these results suggest that the increase of the Young's modulus of the PEG-G8 hydrogels with the increase of strain rates is indeed due to the unfolding of GB1. Therefore, we have included the new experimental results to support our explanation of the strain rate dependent Young's modulus of the PEG-G8 hydrogel.

Revision:

Fig. S3. Tensile test on the pure PEG hydrogel made of four-armed PEG-maleimide and dithiothreitol (DTT). The hydrogel is named as the PEG-DTT hydrogel. **a**, Stress-strain curves for the hydrogel at different strain rates. **b**, Young's Modulus of the hydrogel at different strain rates. (On page 3 of SI)

Fig. S4. Tensile test on the PEG-G8 hydrogel in the presence of 8 M urea. **a**, Stress-strain curves for the PEG-G8 hydrogel with urea at different strain rates. **b**, Young's Modulus of the PEG-G8 hydrogel with urea at different strain rates. (On page 3 of SI)

“Another line of evidence that GB1 domains remained folded in the PAA-G8 hydrogel came from the tensile experiments at varied strain rates (Fig. 2i-k). We did not observe any change of Young's modulus of the PAA and PAA-G8 hydrogels at the strain rates from 5 to 300 mm min⁻¹, indicating that the strain rates were slower than the speed of the polymer chain uncoiling and no rupture of sacrificial bonds or secondary networks occurred during the stretching process^{47,48}. However, in the PEG-G8 hydrogels, the Young's modulus increased greatly with the increase of strain rates, suggesting that GB1 unfolded irreversibly upon stretching, as the PEG hydrogels without G8 cross-linkers or with unfolded G8 cross-linkers showed constant Young's modulus at varied strain rates (Fig. S2 and S3).” (on page 5 of the revised manuscript, last paragraph)

Comment 2:

In the paragraph before figure 3, the authors said “the hydrogel behaved like the PEG-G8 hydrogel”. The authors should explain “behaved” in which way. Low hysteresis? But it's discussing fracture toughness here. Fracture toughness? But the critical strain and fracture energy of PEG-G8 were not provided anywhere in the manuscript. And the critical strain was too low of PEG-G8, it's hard to say any of the PAA-G8 behaved like PEG-G8 in this way. Moreover, the critical strain and fracture energy of PEG-G8 should also be provided.

Response:

We thank the reviewer for this comment. We apologize for the ambiguous claim in the original manuscript. We meant that once the PAA is no longer the percolating phase, the Young's modulus of the PAA-G8 hydrogels became strain rate dependent, similar to that of the PEG-G8 hydrogels (Fig. S7). The reviewer is correct that the critical strain of PEG-G8 was too low compared to that of the PAA-G8 hydrogel. It is not fair to compare their Young's moduli directly. We have revised this sentence in the revised manuscript. Moreover, the Young's modulus, critical strain, and fracture energy of the PEG-G8 hydrogels at varied compositions are now provided in the revised manuscript (Fig. S9).

Revision:

Fig. S7. Tensile test of the PAA-G8 hydrogels (120 mg mL⁻¹). **a**, Stress-strain curves for the hydrogels at different strain rates. **b**, Young's Modulus of the hydrogels at different strain rates. (On page 5 of SI)

Fig. S9. Composition greatly affects the mechanical behaviors of the PEG-G8 hydrogel. **a**, **b**,

Stress-strain curves and the corresponding elastic moduli of the hydrogels of various concentrations of SNAP-G8-SNAP and the same 4-armed-PEG-SH. The samples were unnotched. c, d, Critical strain and fracture energy of notched hydrogels of various concentrations of SNAP-G8-SNAP. (On page 5 of SI)

“The Young’s modulus of the PAA-G8 hydrogels became strain rate dependent, similar to that of the PEG-G8 hydrogels (Fig. S7).” (On page 7 of the revised manuscript)

“The mechanical properties of PEG-G8 hydrogels were also dependent on the composition, despite that the Young’s modulus were consistently higher than that of the PAA-G8 hydrogels at the same SNAP-G8-SNAP concentrations, presumably due to their distinct network structures (Fig. S9). Our results highlight the importance of the components and network structure on the mechanical performance of hydrogels.” (On page 7 of the revised manuscript)

Comment 3:

“It is worth noting that the fracture stress of the PAA-G8 hydrogels was much higher than that of the PAA and PEG-G8 hydrogels”. But in figure 2, the fracture stresses were about 30 for PAA (c&l), 140 for PEG-G8 (d&m), 100 (e) or 80 (n) for PAA-G8. PEG-G8 is the highest.

Response:

We thank the reviewer for this comment. We have now corrected this sentence as “It is worth noting that the fracture stress of the PAA-G8 hydrogels was much higher than that of the PAA hydrogel”. We apologize for not proof-reading the manuscript carefully.

Revision:

“It is worth noting that the fracture stress of the PAA-G8 hydrogels was much higher than that of the PAA hydrogel.” (On page 5 of the revised manuscript)

Comment 4:

The authors used “toughness” and “fracture toughness” in the manuscript, they could have different meaning. Be sure they were clarified. For example, “In contrast, the maximum strain and toughness first increased...”. This description is more likely for intact samples, especially followed by “We also used notched samples...”. But fig. 3c and d were for notched samples. This part is very misleading and confusing. And also for other parts, “fracture toughness” should be used instead of “toughness” if discussing notched samples and anti-tearing properties.

Response:

We thank the reviewer for his/her comments. Following the reviewer’s suggestion, we have distinguished “toughness” and “fracture toughness” clearly in the revised manuscript.

Comment 5:

“Moreover, the diffusion rate increased very slowly with the increase of the strain. Contractively, the crack diffusion rate increased sharply after reaching the threshold. This further suggested that the PAA-G8 hydrogels can significantly prevent fatigue fracture.” The description here is also

confusing. How could these phenomena suggest “the PAA-G8 hydrogels can significantly prevent fatigue fracture”? Did the author want to compare the “slow increase of diffusion rate of PAA-G8 from strain 6 to 8” with “sharply increased diffusion rate of PAA gel after the threshold”? Please state clearly.

Response:

We thank the reviewer for his/her comments. We have corrected this confusing sentence in the revised manuscript.

Revision:

“Moreover, the crack diffusion rate increased very slowly with the increase of strain rates even when the strains were beyond the fracture threshold (i.e. at strains from 6 to 8). In contrast, the crack diffusion rates of the PAA and PEG-G8 hydrogels increased sharply after reaching the threshold. This further suggested that the PAA-G8 hydrogels can significantly prevent fatigue fracture.” (On page 8 of the revised manuscript)

Comment 6:

In figure 4, the unfolding could be easily monitored through fluorescence. A similar experiment stretching intact PAA-G8 gel with ANS would provide more convincing evidence and more insight about the process. And with this consideration, the “(3) the fluorescent labeling experiments” “until the hydrogels approach the fracture limit” mentioned in discussion part was not provided.

Response:

We thank the reviewer for his/her comments. We have performed the ANS labeling experiments on the intact PAA-G8 hydrogel. The new results are included in Supplementary Fig. S11 and Movie 2 in the revised manuscript. The results show that GB1 largely remained folded before the hydrogel reached the fracture limit. A few fluorescent spots at the edge of the hydrogel can be seen during stretching, which may indicate stress-concentration at those areas due to the presence of defects.

Revision:

Fig. S11. Sequential images of stretching an intact PAA-G8 hydrogel ($G8: 100\text{mg mL}^{-1}$) in the presence of an environment sensitive dye, ANS. (On page 7 of SI)

“We also performed the ANS labeling experiments on intact PAA-G8 hydrogel (Fig. S11 and Movie 2). The results show that GB1 largely remained folded before the hydrogel reached the fracture limit. A few fluorescent spots at the edge of the hydrogel can be seen during stretching,

which may indicate stress-concentration at those area due to the presence of defects.” (On page 8 of the revised manuscript)

Comment 7:

The stretching images of notched PEG-G8 gel with ANS in Figure S7 does not show clear precut as for PAA-G8. More representative and persuasive images and video should be attached. Also, stretch results of an intact PAA-G8 gel with ANS are necessary.

Response:

We thank the reviewer for his/her comments. We have now performed the experiment on both the intact PEG-G8 gel and that with a precut. The new results are included in Supplementary Fig. S12, S13 and Movie 3 & 4 in the revised manuscript. The increase of the fluorescent intensity upon stretching indicated that GB1 gradually unfolded in the intact PEG-G8 hydrogel (Fig. S12). In contrast, for the intact PAA-G8 hydrogel, we did not observe obvious increase of fluorescent intensity until the hydrogel was stretched almost to the fracture limit (Fig. S11 and Movie 2). For the notched PEG-G8 hydrogel, the fluorescence at the crack site was brighter than the other part of the hydrogel even at low strains, because the stress at the crack tip was higher. However, the other part of the hydrogel was also considerably lighted up due to the unfolding of GB1.

Revision:

Fig. S12. Sequential images of stretching an intact PEG-G8 hydrogel in the presence of an environment sensitive dye, ANS. The fluorescence of ANS became brighter when it bound with the hydrophobic residues upon GB1 unfolding. (On page 7 of SI)

Fig. S13. Sequential images of stretching a notched PEG-G8 hydrogel in the presence of an environment sensitive dye, ANS. (On page 7 of SI)

Comment 8:

“Based on these data, the extensions of crack per cycle, dc/dN , as a function of the energy release rate, G , are plotted in Fig. 5d”, Please provide in SI how the calculation was carried out according to the experimental results.

Response:

We have provided the calculation in the revised SI.

Revision:

“To record the crack extension, we took photos every 100 cycles (10 cycles for some tests) throughout the test with a digital camera. The photos were post-processed to obtain the crack extension (dc) over the number of loading cycles (dN). Then we can get the extensions of crack per cycle, dc/dN .” (On page 8 of SI)

Comment 9:

In the “Characterization of fatigue fracture of hydrogels” part, what’s the cycled strain for PAA and PEG-G8? The one (5.5) for PAA-G8 already exceeded the critical strain of these two gels. And “Even after 5000 cycles, we did not observe any measurable crack propagation”. But in figure S8c, obvious propagation was demonstrated with 5.5 strain, any explanation?

Response:

We thank the reviewer for his/her comments. The PAA-G8 hydrogel was cycled in between 1 and 5 (not 5.5), and the representative stress-strain curves are shown in Figure 5c. We apologize for not proof-reading the manuscript carefully. The PAA and PEG-G8 hydrogel were also cycled in between 1 and 5. These experimental conditions were included in the revised manuscript.

Revision:

“For all three kind of hydrogels, the strain was cycled between 1 and 5.” (On page 9 of the revised manuscript)

Comment 10:

In discussion, “This is reasonable because some protein domains may be already unfolded prior to reaching the fracture limit”. But in the whole manuscript, “Since the polyprotein cross-linker does not form the percolating phase, they do not experience sufficiently high forces to unfold until the hydrogels approach the fracture limit” is a key point, and suggested by experiment results. Is it suitable to use it as a reason (without evidence and in contradiction with previous arguments) for the difference between theoretical and experimental value? And 272 is not just “a bit” higher than 126. Please provide more reasonable explanation.

Response:

We thank the reviewer for his/her comments. We have removed “some protein domains may be already unfolded prior to reaching the fracture limit”. We apologize that in our original analysis,

we did not consider the geometric arrangement of the G8 cross-linkers. The polyproteins are randomly distributed in the fracture zone and only the polyprotein cross-linkers across the crack tip can be unfolded. The cross-linkers parallel to the crack growth direction did not unfold. Therefore, a prefactor of $\frac{1}{2}$ should be added to the original equation and the fracture threshold is now calculated to be 138 J m^{-2} . See the response to the next comment for calculation details. We have redrawn Figure 5e to illustrate this more clearly. Moreover, we tuned down our claim in the revised manuscript and suggested that the original Lake-Thomas model does not consider energy dissipation (e.g. viscoelasticity, poroelasticity and protein unfolding) of real soft materials and the way we estimated the energy dissipation based on single molecule force spectroscopy data may have certain systematic errors due to the assumption of the force loading rates during crack propagation and the complexity of the network structures¹. The model should be improved in the future to provide quantitative prediction of the fracture threshold.

Revision:

Fig. 5e. Schematic illustration of the unfolding of polyprotein cross-linkers to prevent crack propagation in the PAA-G8 hydrogel but not in the PAA hydrogel. (On Page 10 of the revised manuscript)

“The polyproteins are randomly distributed in the fracture zone and only the cross-linkers perpendicular to the crack growth direction are subjected to stretching forces and unfold (Fig. 5e). The cross-linkers at the parallel positions experience lower strains and do not unfold. By considering these effects, the fracture threshold is calculated to be 138 J m^{-2} , which is close to the experimentally determined value (126 J m^{-2}) (See Fig. S17 and Supporting Information for calculation details). It is worth mentioning that in the original Lake-Thomas model, except for chain scission, other energy dissipation (e.g. viscoelasticity, poroelasticity, and protein unfolding) in real soft materials is not considered. The way we estimated the energy dissipation based on single molecule force spectroscopy data may have certain systematic errors due to the assumption of the strain rates during crack propagation and the complexity of the network structures⁵⁷. Some protein domains may remain folded before the breakage of the cross-linker, if the local strain rate is too fast. The model should be further improved in the future to provide quantitative prediction of the fracture threshold.” (On Page 11 of the revised manuscript)

Comment 11:

In discussion “the calculation further confirm that the polyprotein cross-linkers contribute greatly to the fatigue threshold but little to the toughness”. First, how did the calculation “confirm” it? The analysis is missing. Second, PAA cross-linked by Bis showed both much lower toughness of intact samples (strain energy density, not calculated but obvious from the area under the curve in fig 2c and e) and fracture toughness (fracture energy, ~35 compared with ~800) than PAA-G8. The most difference is the polyprotein cross-linkers. How could it contribute “little to the toughness”?

Response:

We thank the reviewer for his/her comments. We intended to indicate that the polyprotein cross-linkers contribute greatly to the fatigue threshold but do not cause hysteresis. We have corrected this in the revised manuscript. We agree with the reviewer that the calculation is qualitative rather than quantitative. Therefore, we changed “confirm” to “suggest”. The revised calculation detail is provided in SI. The reviewer is correct that the fracture toughness of the PAA-G8 hydrogel is much higher than that of the PAA hydrogel. We have changed “little to the toughness” to “little to the hysteresis”.

Revision:

“the calculation further suggests that the polyprotein cross-linkers contribute greatly to the fatigue threshold but little to the hysteresis”. (On Page 12 of the revised manuscript)

Estimating the fatigue threshold using the Lake-Thomas model (On page 9 of SI)

In the Lake-Thomas theory, the fracture threshold is equal to the number of chains per cross-sectional area multiplied by the energy that is required to break one bridging strand,

$$\Gamma = \sigma W = \frac{1}{2} v_0 R_0 n U,$$

where R_0 is the average end-to-end distance of an elastically active network strand in its undeformed state, v_0 is the number density of such elastically active subchains, n is the average number of repeat units along the bridging strand, and U is the energy that is stored in each repeat unit when the bridging strand breaks. The prefactor of 1/2 comes from the projection of the end-to-end vectors of subchains onto the normal of the crack plane.

Suo and colleagues have recently applied the Lake-Thomas theory to calculated the theoretical fatigue threshold of single-network PAA hydrogels as follows²:

$$\Gamma_0 = \phi_{PAA}^{2/3} b U l n^{1/2},$$

where ϕ_{PAA} is the volume fraction of the polyacrylamide network in the PAA hydrogels, b is the number of bonds per unit volume of the dry polymer, U is the C-C bond energy, l is the length of each monomer unit and n is the number of monomer units in a polyacrylamide chain. For PAA hydrogel, the volume fraction of the polyacrylamide network can be estimated by

$$\phi_{PAA} = \frac{(1-w_{water}) \times \frac{m_{PAA}}{m_{PAA} + m_{Bis}}}{\rho_{AAM}} \times \rho_{gel}.$$

The density of acrylamide ρ_{AAM} is 1.1 g cm^{-3} . The density of the hydrogel is approximately 1 g cm^{-3} . The volume fraction of the polyacrylamide network in the hydrogel is 5.68 vol%. The number of bonds per unit of the dry polymer is estimated by the number of monomers per volume of the dry polymer, $b = A \rho / M = 9.32 \times 10^{27} \text{ m}^{-3}$, where A is the Avogadro number (6.022×10^{23}) and M

is the molecular weight of acrylamide (71.08 g mol^{-1}). The energy of a C-C bond U is $3.3 \times 10^{-19} \text{ J}$. The length of the monomer is estimated by $l=b^{-1/3}=0.475 \text{ nm}$. The molar ratio of the cross-linker bisacrylamide relative to the monomer acrylamide is 0.079%. One cross-link connects two polymer chains, so the number of monomer between two cross-links n is estimated by $n = 1/(2 \times 0.079\%) = 633$. Accordingly, the fatigue thresholds of PAA hydrogel is predicted to be 5.1 J m^{-2} .

In the PAA-G8 hydrogel, we revised the Lake-Thomas theory to consider the energy dissipation from GB1 unfolding. Every GB1 unfolding event leads to a release of potential energy stored by the bridging strand and a change of the end-to-end distance of the network strand. The final energy release comes from the rupture of the bridging strain. Considering the bridging strain made of the PAA chain and eight GB1 domains, the crack energy threshold can be estimated as follows:

$$\Gamma = \sum_{\delta=0}^7 \frac{1}{2} v_0 R (n_a + 3\delta N_{GB1}) U_{uf} + \frac{1}{2} v_0 (R' + 8L_0) (n_a + 3\delta N_{GB1}) U_{break}$$

$$R = \begin{cases} R_0, \delta = 0, \\ R' + \delta L_0, \delta \neq 0, \end{cases}$$

where v_0 the number density, R is the average end-to-end distance of the active subchains, n_a is the number of acrylamide per unit, δ is the number of unfolded GB1, N_{GB1} represents the number of amino acids of GB1, R_0 is the average end-to-end distance of the active subchains in initial, R' is the end-to-end length of the bridge strand before one of GB1 unfolding, U_{uf} is the energy stored in C-C bond at the unfolding force of GB1 and U_{break} is the energy to break C-C bond.

In this equation, the first part corresponds to the energy release due to the unfolding of GB1 domains and the second part correspond to the energy release due to the rupture of the polymer network after the unfolding of all GB1 domains. As the initial concentration of SNAP-G8-SNAP in our hydrogel is 100 mg mL^{-1} , and its molecular weight is $8.9 \times 10^4 \text{ g mol}^{-1}$, with the swelling ratio of 1.6, v_0 is estimated to be $4.2 \times 10^{23} \text{ m}^{-3}$ and R_0 is 28 nm. The number of acrylamide per unit n_a is 1266. U_{uf} is the energy stored in C-C bond at the unfolding force of GB1 (typically 200 pN). Based on the simulation using Worm-Like-Chain model, the potential energy under 200 pN is about 30% of the energy to break a C-C bond ($U_{break}=3.3 \times 10^{-19} \text{ J}$), showed in Fig. S17. Thus U_{uf} is about $0.99 \times 10^{-19} \text{ J}$ per bond. R' is the end-to-end length of the bridge strand before one of GB1 unfolding, which is estimated to be 152 nm (the length per bond 0.12nm multiply n_a). $N_{GB1}=60$ is the number of amino acids of GB1 and $L_0=21.8 \text{ nm}$ is the contour length of GB1. Therefore, the crack energy threshold is estimated to be 138 J m^{-2} .

Fig. S17 Simulation of stretching a polymer to different forces using Worm-Like-Chain model (contour length = 150 nm and persistence length = 0.4 nm). The shaded areas represent the stored potential energy under specific forces.

Some points about the writing:

1. The introduction is quite lengthy and much irrelevant information can be omitted or shortened, such as the evolution of tough hydrogels and examples of folded proteins in natural biological tissues. However, the authors should make it clear why achieving low hysteresis, high stretchability, and high fatigue resistance, concurrently in a single hydrogel is so difficult from a fundamental point of view, and why your design is smart and unique in overcoming this difficulty. Although there are some explanations on the latter in the last paragraph of introduction, it focuses on crack propagation with too much detail and does not cover all three properties.

Response:

We thank the reviewer for this comment. We have rewritten the introduction following the reviewer's suggestion.

Revision:

“The development of soft stretchable materials, including elastomers and gels^{1,2}, enables the fast-growing fields of flexible electronics^{3,4}, tissue engineering scaffolds^{5,6}, and smart drug delivery systems⁷. In many of these applications, high stretchability, fracture toughness and low hysteresis are the prerequisite; yet most hydrogels are too fragile to tolerate cyclic mechanical loads. Single-network hydrogels lack a mechanism to prevent crack propagation and facilitate energy dissipation. Therefore, they are intrinsically weak and prone to fracture. On the other hand, double-network hydrogels are famous for their high stretchability and toughness due to the presence of a sacrificial network that can be fractured to dissipate mechanical energy⁸⁻¹³. However, they inevitably show obvious hysteresis in the stretching-relaxation cycles, making them unsuitable for applications requiring dynamic mechanical loads. Moreover, recent studies have showed that double-network hydrogels are unable to prevent fracture propagation at the strain limit in that the sacrificial network has already been ruptured before reaching the fracture point^{14,15}. As

such, the fracture energy of tough double-network hydrogels is comparable to the intrinsic fracture energy of single-network hydrogels^{14,15}. Recently, Zhao and coworkers developed a way to engineer anti-fatigue fracture hydrogels by inducing crystalline phases to prevent crack propagation¹⁶. Despite that the fatigue threshold was significantly improved, the hysteresis may remain an issue. The stretchability, toughness, hysteresis, and anti-fatigue fracture are all the results of energy dissipation but under different conditions. The seemingly conflict requirements of low hysteresis, high toughness, and fracture resistance make it challenging to design hydrogels combining these mechanical properties.

Unlike synthetic hydrogels, many biological tissues, such as muscle^{17,18} and cartilage¹⁹, show exceptional mechanical properties and can survive under millions of mechanical cycles in their life span. In many of these tissues, nature has evolved a special class of elastomeric proteins made of tandem repeats of folded protein domains to function as cross-linkers for loosely packed proteinaceous fibers²⁰⁻²². These elastomeric proteins can unfold to dissipate mechanical load and quickly refold to recover their original mechanical properties. Inspired by this design, Li and others have pioneered the use of folded protein domains as the building blocks for engineering synthetic hydrogels with tailored mechanical properties²³⁻²⁵. Despite great success in these studies to partially mimic the mechanical response of biological tissues, most engineered hydrogels were still limited by obvious hysteresis, poor stretchability and low fracture thresholds. It remains largely unexplored to engineer stretchable, low hysteresis, and anti-fatigue hydrogels using biomimetic approaches.” (On page 1 and 2 of the revised manuscript)

2. In the introduction, “Therefore, these folded protein domains can efficiently prevent fracture propagation through unfolding without causing additional hysteresis”. This sentence is very confusing. The protein is folded or unfolded? Why unfolding does not cause hysteresis? Although reading the rest of the article can give a clear picture of this work, these summary sentences are very important and should be rewritten.

Response:

We thank the reviewer for this comment. We have rewritten this sentence.

Revision:

“Therefore, the hydrogels show low hysteresis upon stretching. However, these folded protein domains can still be unfolded at the stress-concentrated crack area to efficiently prevent crack propagation, entailing the hydrogels high fatigue resistance.” (On page 2 of the revised manuscript)

Reviewer #3 (Remarks to the Author):

Comments

The manuscript written by Hai Lei and Liang Dong, et al. describes a fascinating design concept of a hydrogel that possesses an ability to decouple the hysteresis-toughness correlation. To achieve this, the authors utilize tandem-repeat proteins (called G8) as the functional cross-linker which works as a shock-absorbable architecture. When forces acting on the polymer network reach a critical value, unfolded G8 units fold to prevent mechanical fatigue and dissipate energy. This concept has already

been reported as mentioned in the manuscript (ref 24, 25), however, the authors newly find that a specific network structure enables the ability to decouple the network elasticity with local mechanical response of the G8 cross-linker. Specifically, the folded G8 cross-linker is embedded in a percolated random coiled polyacrylamide network (called PAA-G8 hydrogel) to separate G8 units from the percolated phase. This unique structure prevents G8 unit folding until the hydrogels approach the fracture limit, resulting in excellent mechanical properties such as high fracture strain (~1100%) with low hysteresis (>5%). To verify the relationships between the network structure and mechanical properties, the authors fabricate another type of hydrogel which contain G8 units in the percolated phase (called PEG-G8 hydrogel). By measuring the mechanical properties and comparing the results of PAA-G8, PEG-G8 and neat PAA hydrogels, the effect of network structure is successfully revealed. Furthermore, the authors establish the method to visualize the protein unfolding during tensile testing by utilizing an environmentally sensitive dye (called ANS). These results agree well with the simulations performed by the finite element analysis method. Finally, fatigue fracture of hydrogels are systematically characterized by cyclic loading/unloading tests. Among them, the PAA-G8 hydrogel exhibits outstanding anti-fatigue properties even after 5000 cycles with a fatigue fracture

threshold of 126 Jm^{-2} . By comparing this value and the theoretically calculated one, the experimental results are a bit lower but reasonable, taking protein folding stability into consideration.

I believe that the main research findings of this paper will be important towards understanding network structures containing tandem-repeat units and designing highly stretchable, low hysteresis and anti-fatigue hydrogels. Moreover, the subject matter of this work is laudable and of interest to various science communities such as soft robotics, flexible sensors and smart wearable devices.

Therefore, I would recommend this paper for publication in Nature Communications; however, several points as indicated below should be addressed by the authors to improve the quality of the article.

Response:

We thank the reviewer for his/her positive comments.

Comment 1:

In the section [Mechanical characterization of the PAA-G8 hydrogels], the authors systematically change the protein cross-linker concentration. When the concentration exceeds a certain point, the authors mentioned that PAA is no longer the percolated phase (some folded proteins were involved in the main network). Could you verify this by using a visual method as the authors established?

Response:

We thank the reviewer for this comment. Following the reviewer's suggestion, we have performed the stretching experiments on PAA-G8 hydrogels with varied G8 concentrations in the presence of ANS. The videos and the figures are shown in Fig. S11, S14, S15 and Movie 2, 5, 6. Our results clearly showed that when PAA was no longer the percolating phase, the hydrogel was lighted up upon stretching due to the unfolding of GB1. In contrast, at low G8 concentrations, the hydrogels

remained dim. These results further suggest that having PAA as the percolating phase is critical for achieving low hysteresis, high stretchability, and high fatigue resistance.

Revision:

Fig. S11. Sequential images of stretching an intact PAA-G8 hydrogel (G8: 100 mg mL⁻¹) in the presence of an environment sensitive dye, ANS. (Movie 2) (On page 7 of SI)

Fig. S14. Sequential images of stretching an intact PAA-G8 hydrogel (G8: 120 mg mL⁻¹) in the presence of an environment sensitive dye, ANS. In this hydrogel, the crack started to propagate from the left side, presumably due to the presence of defects there. (Movie 5) (On page 7 of SI)

Fig. S15. Sequential images of stretching an intact PAA-G8 hydrogel (G8: 80 mg mL⁻¹) in the presence of an environment sensitive dye, ANS. In this hydrogel, the crack started to propagate from the top side, presumably due to the presence of defects there. (Movie 6) (On page 8 of SI)

“Furthermore, we have performed the stretching experiments on the PAA-G8 hydrogels with varied G8 concentrations in the presence of ANS (Fig. S14, S15 and Movie 5, 6). Our results clearly showed that at the G8 concentration of 120 mg mL⁻¹, PAA was no longer the percolating phase and the hydrogel was lighted up upon stretching due to the unfolding of GB1. In contrast, at a lower G8 concentration of 80 mg mL⁻¹, PAA remained as the percolating phase and the whole hydrogel, except for the crack area, kept dim. These results further suggest that having PAA as the percolating phase is critical for achieving low hysteresis, high stretchability, and high fatigue resistance.” (On page 8 of the revised manuscript)

Comment 2:

In Fig 4, we can clearly observe fluorescence indicating protein folding around the crack-propagation site. If the author do the same experiment without precut, how does the fluorescent intensity change? You mentioned that G8 units in PAA-G8 gel do not experience sufficiently high force to unfold until the hydrogel approaches the fracture limit. Could you verify this by your method? Also, how about the intensity change in the case of PEG-G8?

Response:

We thank the reviewer for his/her comments. We have performed the fluorescent labeling experiments on the intact PAA-G8 and PEG-G8 hydrogels (in Fig. S11, S12 and Movie 2, 3). The results showed that the intact PAA-G8 hydrogels mainly remained dark before being stretched to the fracture limit. There were only a few fluorescent spots showing up on the PAA-G8 hydrogel during stretching due to the presence of defects. In contrast, the PEG-G8 hydrogel was gradually lighted up during stretching and the fluorescence reached the maximum at the strain of 2.4, which indicates the unfolding of GB1.

Revision:

Fig. S11. Sequential images of stretching an intact PAA-G8 hydrogel (G8: 100mg mL^{-1}) in the presence of an environment sensitive dye, ANS. (On page 7 of SI)

Fig. S12. Sequential images of stretching an intact PEG-G8 hydrogel in the presence of an environment sensitive dye, ANS. The fluorescence of ANS became brighter when it bound with the hydrophobic residues upon GB1 unfolding. (On page 7 of SI)

Comment 3:

To stabilize the protein unit, the hydrogels were swollen in PBS. I'm curious about the environmental stability of G8 cross-linker. Does it work in pure water? Does it work even after drying then reswelling? If the G8 cross-linker is environmentally durable, it might be good for

applications such as wearable devices and soft robotics.

Response:

We thank the reviewer for his/her comments. We have placed the hydrogel in pure water. Due to the high stability of GB1, it showed similar performance in pure water as that in PBS. It also worked after drying and then rehydration. We have included these results in the revised manuscript and indicated that because the G8 cross-linker is environmental durable, the hydrogels might be good for applications such as wearable devices and soft robotics.

Revision:

Fig. S18. Photos of the hydrogel swollen in pure water (left), after drying (middle) and reswelling (right). (On page 10 of SI)

Fig. S19. The stress-strain curves for the PAA-G8 hydrogel in PBS (black line), pure water (green line) and reswelling in pure water (orange line). The curves were horizontally offset for clarity. (On page 11 of SI)

“It is worth mentioning that the G8 cross-linker is stable in pure water and tolerable to dehydration. The hydrogel can be dehydrated and rehydrated in water without causing significant changes to the mechanical properties (Fig. S18 and S19). We anticipate that these hydrogels can find broad applications in soft robotics, flexible sensors, and smart wearable devices, where the materials are routinely subjected to multiple load/unload cycles.” (On page 12 of the revised manuscript)

Comment 4:

Minor points:

In Fig.2(c-e),(f-h),(i-k),(l-n), it is little bit hard to tell which date represents which gel. Could you put a label or subheading in the figure?

Response:

We thank the reviewer for his/her comments. We have revised the figure following the reviewer's suggestion.

Fig. 2. The bulk mechanical properties of the designed hydrogels. **a**, A strip of the undeformed PAA-G8 gel was fixed to two rigid clamps on the top and the bottom. The hydrogel was stretched to 11 times its initial length in a tensile test. **b**, A hydrogel with a pre-cut notch (yellow line) was stretched to 5.5 times its initial length without fracture. **c-e**, The stress-strain curves for the PAA, PEG-G8 and PAA-G8 hydrogels until break. **f-h**, Representative stretching-relaxation curves for the PAA, PEG-G8 and PAA-G8 hydrogels. The curves are horizontally offset for clarity. The final strains are shown on the curves. Insets show the superposition of the stretching-relaxation curves at different strains. **i-k**, Stress-strain curves for the PAA, PEG-G8 and PAA-G8 hydrogels at different initial deformation rates. **l-n**, Fracture stress and Young's Modulus of the PAA, PEG-G8 and PAA-G8 hydrogels at different deformation rates. (On page 6 of the revised manuscript)

Reference

- 1 Shu, W. et al. Quantitative Adjustment to the molecular Energy Parameter in the Lake-Thomas Theory of Polymer Fracture Energy. *Macromolecules* **52**, 2772-2777 (2019).
- 2 Zhang, W. et al. Fracture Toughness and Fatigue Threshold of Tough Hydrogels. *ACS Macro Lett.* **8**, 17-23 (2019).

REVIEWERS' COMMENTS:

Reviewer #1 (Remarks to the Author):

The authors fully addressed the questions raised by the reviewer, and the paper is now recommended for publication as it is in Nature Communication.

Reviewer #2 (Remarks to the Author):

The authors have responded to all the comments, addressed the concerns, and clarified the confusing parts in the revised version. The manuscript has been well improved and acceptable for publication. But there is still one question not clear. In reply to reviewer 2 Comment 8, the measurement and calculation of dc/dN has to be provided. In addition, the measurement and calculation of energy release rate should also be provided.

Reviewer #3 (Remarks to the Author):

Authors addressed the review comments politely. The manuscript has been much improved and is in a nice condition now. This paper is an important contribution and I recommend that it be accepted for publication in Nature Communications.

Point by point response

Reviewer #2 (Remarks to the Author):

The authors have responded to all the comments, addressed the concerns, and clarified the confusing parts in the revised version. The manuscript has been well improved and acceptable for publication. But there is still one question not clear. In reply to reviewer 2 Comment 8, the measurement and calculation of dc/dN has to be provided. In addition, the measurement and calculation of energy release rate should also be provided.

Response:

We thank the review for this comment. The calculation details are now provided in the Supplementary Information.

The extension of crack per cycle, dc/dN , is measured from the images of the undeformed hydrogels before and after the N^{th} cycle. $dc/dN = L_N - L_{N-1}$, where L_N and L_{N-1} are the length of the crack after the N^{th} and $(N-1)^{\text{th}}$ stretching-relaxation cycle, respectively. To record the crack extension, we took photos every 100 cycles (10 cycles for some tests) throughout the test with a digital camera. The photos were post-processed to obtain the crack extension (dc) over the number of loading cycles (dN). Then we can get the extensions of crack per cycle, dc/dN .

The energy release rate G of the notched hydrogel takes the following form:

$$G = HW(\lambda_{max})$$

Where H is the distance between the two grippers of the tensile tester when the notched hydrogel is undeformed; $W(\lambda_{max})$ is the energy per volume of the uncut sample while stretched; and λ_{max} is the critical strain in which the crack growth rate is the lowest obtained in experiments. The energy density $W(\lambda_{max})$ is obtained by integrating the area below the stress-strain curve of the unnotched sample.